# Efficacy and safety of saroglitazar for the management of dyslipidemia: A systematic review and meta-analysis of interventional studies

**Manik Chhabra**[1,2], **Kota Vidyasagar**[3], **Sai Krishna Gudi**[4], **Jatin Sharma**[5], **Rishabh Sharma**[1], **Muhammed Rashid**[6]*

1 Department of Pharmacy Practice, Indo-Soviet Friendship College of Pharmacy, Moga, Punjab, India,
2 Rady Faculty of Health Sciences, Department of Pharmacology and Therapeutics, Max Rady College of Medicine, University of Manitoba, Winnipeg, MB, Canada, 3 Department of Pharmacy, University College of Pharmaceutical Sciences, Kakatiya University, Warangal, Telangana, India, 4 Rady Faculty of Health Sciences, College of Pharmacy, University of Manitoba, Winnipeg, MB, Canada, 5 Department of Pharmacology, All India Institute of Medical Sciences, New Delhi, Delhi, India, 6 Department of Pharmacy Practice, Sri Adichunchanagiri College of Pharmacy, Adichunchanagiri University, BG Nagara, Karnataka, India

* muhammedrashid2@gmail.com

**Data Availability Statement:** All relevant data are within the paper and its Supporting Information files.

## Abstract

### Background and objective

Saroglitazar is a newer antidiabetic agent approved to manage dyslipidemia. The objective is tevaluate the efficacy and safety profiles of saroglitazar in patients with dyslipidemia.

### Methods

A systematic search was conducted using PubMed, Cochrane Library, Scopus, and Google Scholar from the inception until January 2022. Interventional studies comparing the anti-hyperlipidaemic effect and safety of saroglitazar with or without a control group(s) were included. The efficacy of saroglitazar was assessed concerning its effect on total cholesterol, low density lipoprotein (LDL) and high density lipoprotein (HDL)-cholesterol, triglycerides, fasting plasma glucose, and non-HDL cholesterol. The effects on serum creatinine levels, bodyweight reduction, alanine aminotransferase and aspartate aminotransferase were considered to be safety endpoint.The Cochrane risk of bias assessment tool was used to assess the methodological quality of the included studies.

### Results

A total of six studies with 581 adults with a mean age ranging from 40.2 to 62.6 years were included in this study. A significant decrease in low-density lipoprotein cholesterol was observed with saroglitazar 4 mg therapy compared to saroglitazar 2 mg [standardized mean difference (SMD): −0.23 mg/dL, 95% CI: −0.47 to 0.00; p = 0.05; 2 studies], and control [SMD: −0.36 mg/dL, 95% CI −0.59 to -0.12; p = 0.0026; 3 studies]. Also, a significant decrease in the total cholesterol was observed with saroglitazar 4 mg therapy compared to

**Funding:** The author(s) received no specific funding for this work.

**Competing interests:** The authors have declared that no competing interests exist.

saroglitazar 2 mg [SMD − 0.28 mg/dL, 95% CI: − 0.52 to -0.04; p < 0.01; 2 studies], and control [SMD − 0.49 mg/dL, 95% CI: − 0.72 to -0.26; p < 0.0001; 3 studies]. Saroglitazar was not associated with adverse effects such as increase in serum creatinine levels, alanine aminotransferase and aspartate aminotransferase and bodyweight reduction.

## Conclusion

Saroglitazar appeared to be an effective and safer therapeutic option for improving dyslipidemia in patients. However, comparative studies of saroglitazar with the other pharmacological agents are warranted.

## Introduction

According to the atlas of the International Diabetes Federation (IDF), globally, 463 million individuals have diabetes, and it is expected to reach up to 578 million by 2030 and 700 million by 2045 [1]. In India, around 77 million individuals live with type-2 diabetes (T2D) as of 2019, and >4 million people aged between 20–79 years have died from diabetes-related causes [1]. Cardiovascular diseases (CVD) are the primary cause of morbidity and mortality among individuals with T2D [2], where cardiovascular risk tends to be primarily influenced by dyslipidemia [3]. The anti-hyperlipidaemic agents have been studied for decades, and evidence supports their cardiovascular benefit in selected patients with the presence or absence of T2D [4]. Diabetic dyslipidemia is a type of lipoprotein dysfunction characterized by decreased high-density lipoprotein levels, an increase in triglyceride levels, and an increase in low-density lipoprotein (LDL) particles [5]. Approximately around 70% of the patients who have T2D are likely to develop dyslipidemia [6].

Statins, fibrates, cholesterol absorption inhibitors, niacin, bile acid sequestrants, proprotein convertase subtilisin/kexin type-9 (PCSK-9) inhibitors are the available therapeutic options to treat dyslipidemia [5]. However, the use of statins and fibrates is associated with myopathy [7], where niacin is usually not recommended due to the risk of new-onset of diabetes [8], bleeding, and infections [9], and PCSK-9 inhibitors are more expensive and cause injection site reactions. Hence, there is an immense need for newer medications that can potentially target both dyslipidemias as well as T2D [10]. In patients with T2D, peroxisome proliferator-activated receptor alpha (PPAR-α) agonist improves lipid profile, whereas PPAR-γ agonist improves glucose profile [11].

Dual PPAR α/γ agonists have multiple actions, where they act by improving both lipids as well as glucose profiles [12,13]. Saroglitazar [(S)-a-ethoxy-4-{2-[2-methyl-5-(4-methylthio) phenyl)]-1H-pyrrol-1-yl]- ethoxy)-benzenepropanoic acid magnesium salt] is a novel PPAR α/γ agonist synthesized in India by Zydus Cadila (trade name, Lipaglycn), and is approved by the Drug Controller General of India (DCGI) for the treatment of diabetic dyslipidemia and hypertriglyceridemia [14]. Saroglitazar is not associated with weight gain and edema [13], and thus it is considered a safer medication, with fewer adverse effects such as gastritis, asthenia, and pyrexia [15]. There is a lack of pooled evidence on the safety and efficacy of saroglitazar for the treatment of dyslipidemia. With this background, we conducted a systematic review and meta-analysis aimed to examine the efficacy and safety of saroglitazar for managing dyslipidemia.

## Methods

The current systematic literature review was performed and reported according to the Preferred Reporting Items for Systematic Reviews and Meta-Analyses (PRISMA) guidelines [16].

### Eligibility criteria for included studies

The research question for our systematic review is "What is the safety and efficacy of saroglitazar for the management of hypercholesterolemia?" The research question was broken down into PICOS (population, intervention, comparison, outcome, and study design) format. Population (P) comprises adult patients with hypercholesterolemia with or without diabetes. Intervention (I) consists of saroglitazar in any of its effective doses. Comparator (C) was any comparator such as different doses, control, or another drug as per the author's discretion. Outcomes (O) considered were efficacy and safety of Saroglitazar. The effect of saroglitazar on total cholesterol, LDL-cholesterol, triglycerides, high-density lipoprotein (HDL)-cholesterol, non-HDL cholesterol, and fasting plasma glucose is considered to be efficacy outcomes. While the reduction in Serum Creatinine, Alanine Transaminases, Aspartate Transaminases, and body weight considered to be safety outcomes. In study design (S), we included all types of interventional studies; randomized, and non-randomized trials. Any studies satisfying the above specified PICOS were included in the systematic review. Reviews, observational and descriptive studies, editorials, commentaries, and conference proceedings were excluded.

### Data sources and literature search

A comprehensive literature search was performed using PubMed, Scopus, the Cochrane Library, and Google Scholar from the inception to January 2022. The initial search was performed in April 2019 and was updated in January 2022. We employed all the MesH terms and keywords related to "Saroglitazar" and "hypercholesterolemia" obtained from the databases and previously published studies. In addition to this, all the references of the included studies and a snowball search through Google were performed to search for any additional relevant articles. A detailed search strategy for each database is provided in S1 File.

### Study selection and data extraction

The titles and abstracts of retrieved studies were screened, followed by the full text under the predefined inclusion and exclusion criteria. The highly irrelevant studies only were excluded during the title/abstract screening. Two independent reviewers performed the title and abstract screening, followed by full-text screening. A well-defined data extraction sheet was employed for the data abstraction, including the information about studies' characteristics, participants, interventions, comparator, and outcomes. Two independent researchers were involved in study selection and data extraction, disagreements were resolved through consensus or by a discussion with a third reviewer.

### Risk of bias and quality assessment

The Cochrane risk of bias assessment tool was used to assess the methodological quality of the included studies [17]. The quality of the included single-arm trials was evaluated by the National Institute of Health (NIH) Checklist. It consists of 12 items that identify the methodological features that are based on existing study design and study-reporting guidelines. Each item carries one point and scores of 0–4, 4–8, and 9–12 were considered poor, fair, and good-quality studies, respectively. Two independent reviewers evaluated the risk of bias and quality

of included studies and resolved any disagreements through consensus or by a discussion with a third reviewer.

## Data synthesis

The analyses were performed with R software (R version 4.1.2) using a meta-package. Changes in continuous outcomes were calculated for every included study arm by subtracting the value at baseline from the value after the intervention. All the efficacy estimates were expressed as standardized mean difference (SMD) or absolute weighted mean change (MRAW) and 95% confidence interval (CI) from baseline. According to the Cochrane handbook, standard deviations (SD) were calculated from the standard error or 95% CI for a systematic review of interventions [17]. The Higgins I2 statistics and Cochran's Q test were used to assess the potential statistical heterogeneity among trials. The meta-analysis was conducted using a fixed-effect model (using inverse-variance) or a random-effect model (DerSimonian–Laird method) based on low heterogeneity (<50%) or high heterogeneity (>50%). Due to less number of included studies (<10), it was not feasible to examine the publication bias through a funnel plot, Egger's and Begg's test [17].

## Results

A total of 272 citations were identified by electronic search of databases, and 18 citations were identified through hand search. Finally, 276 citations were screened after removing 14 duplicates, while 250 studies were excluded during first-pass screening after reviewing the titles and abstracts. Full texts of 26 citations included during the first pass screening were downloaded for the second pass screening. Lastly, 6 articles met our inclusion criteria. PRISMA flowchart for the study selection is described in Fig 1.

### Characteristics of the included studies

We included 6 studies (4 randomised controlled trials [12,18–20] and 2 single-arm trials [21,22]. Treatment follow-up of the included studies ranged from 12 weeks [12,18,20,22] to 24 weeks [19,21]. All included studies were carried out in India, where four were multicentre trials [12,18–20] and two were single-center trials [21,22]. The detailed characteristics of included studies are presented in Table 1.

### Treatment and outcome characteristsics

There was a total of 581 participants from six studies, of whom the majority of them were females (302 [52%]). All the studies have included adult participants with a mean age of 40.2 [20] to 62.6 years [22]. Of six included studies, two studies [12,19] have compared the effect of saroglitazar 4mg to saroglitazar 2mg and control. The details about treatment and outcome characteristics of included studies are presented in Table 2.

### Quality assessment

Risk of bias of the included studies was assessed using the Cochrane risk of bias assessment tool. Bias related to randomization was low among three (75%) studies, while it appeared to have a higher level of bias with the remaining one study. Bias related to the allocation concealment was found to be low (n = 3; 75%) and moderate (n = 1; 25%) among the studies, respectively. Information about blinding participants and assessors was mentioned in three studies. As per the NIH checklist, both Bhosle D et al., 2018 and Deshpande A et al., 2016 trials did not report the information related to the masking of patients or investigators and did not report

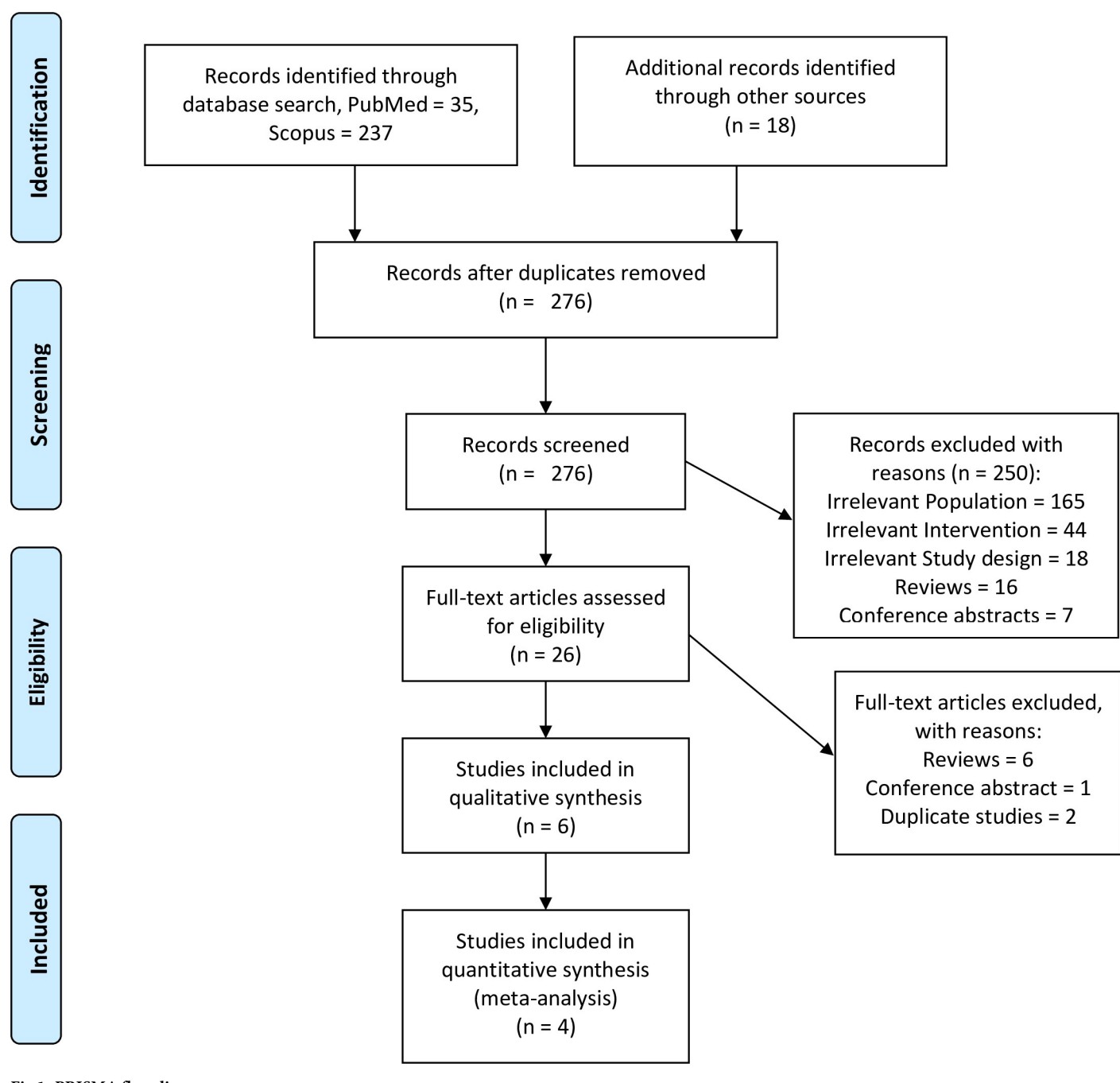

**Fig 1. PRISMA flow diagram.**

the sufficient details of the group-level interventions and individual-level outcome efforts. But overall, two trials were rated as good quality as per rating of NIH checklist. Detailed information on overall risk of bias in individual studies is provided in in Figs 2 and 3 and S1 Table.

## Efficacy of saroglitazar

**Total cholesterol.** Two studies [12,19] compared the saroglitazar 4 mg with 2 mg per day, where there were 138 patients each in the saroglitazar 4 mg/day and 2 mg/day groups that

**Table 1. Characteristics of included studies and subjects.**

| Study ID | Reg No | Country (State/Sites) | Study design | Study duration | Number of participants | Lost to follow up | Age (years) Mean (SD) | Gender N (%) |
|---|---|---|---|---|---|---|---|---|
| Bhosle D et al., 2018 [21] | CTRI/2016/03/006778 | India (Maharashtra) | Prospective, single centre, single arm study | NR | 40 | 0 | 48.15 (7.53) | Male: 28 (70) Female: 12 (30) |
| Rastogi A et al. 2020 [18] | CTRI/2015/06/005845 | India (Chandigarh and Mumbai) | Prospective, Multi centre, double blinded RCT | 12 weeks | Saroglitazar (4mg): 15 | 2 | 53.1 (8.8) | Male: 7 (47) Female: 8 (53) |
| | | | | | Placebo: 15 | 0 | 54.9 (7.8) | Male: 9 (60) Female: 6 (40) |
| Pai V et al., 2014 [19] | Phase III/CTRI/2009/091/000527 | India (Pune, Chennai, Kolkata, Chandigarh and Bangalore) | Prospective, Multi centre, double blind RCT | 24 weeks | Saroglitazor (2mg): 41 | 4 | 48.9 (8.98) | Male: 26 (63) Female: 15 (37) |
| | | | | 24 weeks | Saroglitazor (4mg): 41 | 2 | 47.3 (9.10) | Male: 25 (61) Female: 16 (39) |
| | | | | 24 weeks | Pioglitazone (45mg): 40 | 5 | 49.9.1 (10.98) | Male: 24 (60) Female: 16 (40) |
| Deshpande A et al., 2016 [20] | Phase II/CTRI/2010/091/000107 | India (2 sites) | Prospective, multi-centric, open-label, single arm exploratory study | 12 weeks | 50 | 1 | 40.26 (7.13) | Male: 32 (64) Female: 18 (36) |
| Ghosh A et al., 2016 [22] | Phase IV/CTRI/2014/10/005131 | India (West Bengal) | Prospective, randomized, open labelled, parallel group phase IV clinical trial | 12 weeks | Metformin (1000 mg/day) + Fenofibrate (160 mg/day): 18 | 0 | 58.1 | NR |
| | | | | | Metformin (1000 mg/day) + Saroglitazar (4 mg/day): 19 | 1 | 62.6 | NR |
| Jani RH et al., 2014 [12] | Phase III/CTRI/2009/091/000533 | India (29 centres) | Multicenter, prospective, randomized, double-blind, placebo controlled, interventional, Phase III study | 12 weeks for efficacy, 24 weeks for efficacy | Saroglitazar (2 mg): 100 | 6 | 50.4 (9.01) | Male: 39 (39) Female: 61 (61) |
| | | | | | Saroglitazar (4 mg): 99 | 6 | 51.2 (8.66) | Male: 43 (43.4) Female: 56 (56.6) |
| | | | | | Placebo: 102 | 6 | 49.8 (9.95) | Male: 47 (46.1) Female: 55 (53.9) |

CTRI: Clinical Trial registry of India; NR: Not reported; RCT: Randomized Controlled Trial; SD: Standard deviation

resulted in a standardized absolute mean difference (SMD) of -0.28 **mg/dL,** on total cholesterol (95% CI -0.52 to -0.04), p = 0.01 (Fig 4). This demonstrates a significant decrease in the total cholesterol among the saroglitazar 4 mg/day group compared to saroglitazar 2 mg/day group in patients with diabetes-related hyperlipidaemia (Table 3). There was no statistically significant heterogeneity among included studies (P = 0.29; $I^2$ = 12%). Besides, three studies [12,18,19] compared the saroglitazar 4 mg with control, where there were 148 patients in saroglitazar 4 mg/day group and 144 patients in the control group. The results showed an SMD of -0.49 **mg/dL,** (95% CI -0.72 to -0.26), P<0.0001 (Fig 5). This demonstrates a significant decrease in the total cholesterol among the saroglitazar 4 mg/day group compared to the

**Table 2.** Treatment and outcome characteristics.

| Study ID | Inclusion criteria/Diagnosis | Exclusion Criteria | Intervention/Comparator/Duration of treatment | Endpoint | Efficacy and Safety Outcome observed | Key Conclusion |
|---|---|---|---|---|---|---|
| Bhosle D et al., 2018 [21] | • Pre-diabetes and dyslipidaemia<br>• Age: 20–60 years<br>• Baseline HbA1c 5.7–6.4% and dyslipidemia (Total cholesterol > 200mg/dl, LDL-C > 130 mg/dl, triglycerides > 150 mg/dl and HDL< 40 mg/dl) | • Type I DM, Type II DM, secondary hypertension, bronchial asthma, chronic obstructive pulmonary disease or any other respiratory disorders and any hepatic or renal diseases<br>• On-going medications affecting blood glucose or lipids were excluded | Saroglitazar 4mg once daily/ No comparator/ Int: 24 weeks | **Efficacy**<br>Primary outcome: Change in serum triglycerides; Secondary outcome: Changes in other lipid parameters such as HbA1c, serum total cholesterol, serum LDL-C, serum HDL-C and Non HDL-C levels<br>**Safety:**<br>RFT, LFT and ECG Changes following treatment | **Efficacy**<br>Significant reduction in serum triglycerides HbA1c, serum total cholesterol, serum LDL-C, serum HDL-C and Non HDL-C levels<br>**Safety:**<br>No serious adverse event | Saroglitazar is safe and effective in prediabetes with dyslipidaemia |
| Rastogi A et al. 2020 [18] | • Age: 18–65 years<br>• T2DM on stable dose of metformin (≥ 3 months)<br>• BMI > 25 and < 45 kg/m2<br>• FPG ≤ 240 mg/dL,<br>HbA1c ≥ 6.5% (≥ 48 mmol/mol) and ≤ 8.5% (≤ 69 mmol/mol), TG > 150 and < 300 mg/dL, HDL < 50 mg/dL in women and < 40 mg/ dL in men and creatinine clearance > 60 mL/min | • Treatment with glitazones/ glitazars, insulin, steroid and hormone replacement therapies in last 3 months<br>• History of acute or chronic metabolic acidosis, including diabetic ketoacidosis, abnormal thyroid-stimulating hormone, alanine aminotransferase ≥ 2.5 times the upper limit of normal, congestive heart failure and blood pressure > 160/100 mmHg | Saroglitazar 4mg once daily/12 weeks Placebo/ | **Efficacy**<br>Primary outcome: Change in plasma TG area under the curve (AUC) oral fat tolerance test; Secondary outcomes: Change in AUC of apolipoportein B48 and B100 concentration, TG-AUC, TG, TC, LDL, HDL, FPG, HbA1c<br>**Safety:** Adverse events | **Efficacy**<br>Reduction in TG-AUC, TG, TC, LDL, HDL, FPG, HbA1c<br>**Safety:**<br>Incidence of adverse events and severity of Adverse events | Saroglitazar significantly improved postprandial TGs in people with diabetic dyslipidemia |
| Pai V et al., 2014 [19] | • Age: 18 to 65 years<br>• BMI > 23 kg/m2),<br>hypertriglyceridemia (fasting TG > 200 to 400 mg/dL)<br>• History of T2DM (HbA1c >7% to 9%), and receiving either a sulphonylurea, metformin, or both treatments for at least 3 months. | • Using insulin, glitazone or glitazar, or medications with a lipid-lowering agent in the past 2 weeks<br>• History of cardiac abnormalities, thyroid dysfunction, hepatic dysfunction, gall stones, renal dysfunction, myopathies, active muscle diseases, ketonuria, concurrent serious illness such as severe infections (tuberculosis, HIV), malignancy, alcohol or drug abuse, allergy or intolerance to the study medications, or their excipients and participation in any other clinical trial in past 3 months. | Saroglitazar (2 or 4mg) once daily/ Pioglitazone (45 mg) once daily 24 weeks | **Efficacy**<br>Percentage change in triglyceride form baseline; Secondary endpoint: Change in lipid profile and fasting plasma glucose at week 24<br>**Safety**<br>Adverse events | **Efficacy**<br>Reduction in TG, TC, LDL, HDL, FPG, HbA1c<br>**Safety**<br>Incidence of adverse events and severity of Adverse events | Saroglitazar effective and safe for improving hypertriglyceridemia in patients with T2DM |

(Continued)

Table 2. (Continued)

| Study ID | Inclusion criteria/Diagnosis | Exclusion Criteria | Intervention/Comparator/Duration of treatment | Endpoint | Efficacy and Safety Outcome observed | Key Conclusion |
|---|---|---|---|---|---|---|
| Deshpande A et al., 2016 [20] | • Age: 18–65 years • Diagnosis of HIV1 and on HAART for at least 18 months; on stable ART regimen for at least 8 weeks prior to inclusion in the study and ART regimen not expected to change in next 3 months • Patient clinically diagnosed as HIV lipodystrophy (at least 1 moderate or severe lipodystrophy feature identified by doctor and patient, except isolated abdominal obesity) • Triglycerides level >200 to 500 mg/dL; CD4 count of >50/mm3 and patient who had given written informed consent for participation in the trial | • Patients on insulin and/or glitazone/glitazar therapy • Pregnancy and lactation • History of gall stones, cardiac failure, alcohol and/or drug abuse; history of allergy, sensitivity or intolerance to the study drug and its formulation ingredients; active opportunistic infection in last three months • History of malignancy or active neoplasm; any active hormonal disease and/or hormonal treatment that could affect the outcomes of interest such as clinically overt hypo/hyperthyroidism, hypogonadism, hypercortisolism, or treatment with steroids or growth hormone • Haemoglobin below 9 g/dL or total leucocyte count below 1000/mm3 or platelet count below 50,000/mm3; history of myopathies or evidence of active muscle diseases or CPK ≥10 times upper limits of normal (ULN) • History of active liver disease or hepatic dysfunction demonstrated by aspartate aminotransferase (AST) and alanine aminotransferase (ALT) ≥2.5 times of upper limits of normal or bilirubin more than 2 times UNL • Renal dysfunction (serum creatinine >2 mg/dL) and participated in any other clinical trial in past 3 months | Saroglitazar 4 mg Dialy in the morning/ No comparator/ 12 week | **Efficacy** Primary outcome: Percent change in triglyceride levels from baseline to Week 6 and Week 12. Secondary efficacy endpoints were assessment of LDL, VLDL, HDL, Non HDL cholesterol, total cholesterol, Apo A1, Apo B, and C-peptide and fasting insulin for HOMA beta and HOMA IR from baseline to week 6 and week 12 **Safety:** Adverse events | **Efficacy** Significant reduction in TG, VLDL, TC, Increase in HDL, Apo A1,and Apo B, C-peptide, fasting insulin levels, HOMA of beta cell function for C-peptide, HOMA of insulin resistance for C-peptide **Safety:** AEs was less than 10%. Adverse events from the gastrointestinal disorder system organ class (SOC) (constipation and abdominal pain) | Saroglitazar was effective in changing lipid profile and HIV associated lipodystrophy with minimal side effects |
| Ghosh A et al., 2016 [22] | • Age: 18–70 years • Newly diagnosed cases of diabetic dyslipidaemia with plasma triglyceride level ≥ 150 mg/dl and HbA1C ≥ 6.5 and ≤ 8 • No use of any hypolipidaemic agent within last six months • Patients were receiving only metformin 1000 mg per day | • Pregnant or lactating woman • Fasting plasma glucose (FPG) > 250mg/dl, post-prandial plasma glucose (PPPG) > 350mg/dl, LDL-C > 130 mg/dl • Co-morbid cardiovascular, renal and psychiatric complications • Co-administration of drugs that were likely to interact with saroglitazar, fenofibrate or metformin and that are likely to alter lipid profile and glycaemic status | Metformin SR 1000 mg/day and saroglitazar 4 mg per day/ Metformin SR 1000 mg/ day and fenofibrate 160 mg per day/12 weeks | **Efficacy** Primary outcome: Change in TG at week 12 visit. Secondary outcome: Changes in HbA1C, HDL-C, LDL-C and Total cholesterol (TC) at week 12 and FPG, PPPG at weeks 4, 8, 12 visits **Safety** Body weight and incidence of adverse events | **Efficacy** Significant decrease in TG and HbA1C levels were observed in group B compared to group A at week 12 FPG and PPPG were significantly reduced in group B compared to group A at every interval Inter group analysis did not show any statistically significant change in body weight, LDL-C and HDL-C at week 12 | Saroglitazar was better effective than fenofibrate in lipidemic control when combining with metformin |

*(Continued)*

**Table 2.** (Continued)

| Study ID | Inclusion criteria/Diagnosis | Exclusion Criteria | Intervention/Comparator/Duration of treatment | Endpoint | Efficacy and Safety Outcome observed | Key Conclusion |
|---|---|---|---|---|---|---|
| Jani RH et al 2014. [12] | • Age: 18–65 years<br>• T2DM Diagnosis<br>• Previous treatment with a maximum of two oral hypoglycemic Agents<br>• Low-density lipoprotein cholesterol (LDL-C) level >100 mg/dL; TG level >200 and <500 mg/dL; body mass index >23 kg/m2; and treatment with atorvastatin 10mg for at least 4 weeks | • History of greater than 5% weight loss in the preceding 6 months, unstable angina, acute myocardial infarction in the preceding 3 months, heart failure classified as New York Heart Association Class III-IV<br>• Uncontrolled hypertension, clinically significant edema, thyroid disorder, gallstones, impaired liver (aspartate aminotransferase and alanine aminotransferase ‡2.5 times the upper normal limit [UNL] or bilirubin ‡2 times the UNL) o renal (serum creatinine >1.2 mg/dL) function, ketonuria, myopathies or active muscle diseases (creatinine phosphokinase ‡10 times UNL), severe illness such as tuberculosis, human immunodeficiency infection, malignancy, alcohol and/or drug abuse, allergy, sensitivity or intolerance to the study drugs and their formulation ingredients, and participation in any other clinical trial in the preceding 3 months at the time of enrolment<br>• Treatment within 4 weeks of the run-in period with oral antidiabetes drugs from the glitazone (e.g., pioglitazone or rosiglitazone) and glitazar (investigational products) classes, insulin, lipid-modifying therapy (e.g., fenofibrate) other than atorvastatin 10 mg, thyroid-modulating drug, or anti-inflammatory drugs<br>• Pregnant or breast feeding or had initiated hormonal treatment (e.g., hormonal contraceptive, hormone replacement therapy) in the preceding 3 month | Saroglitazar 2 mg/day or Saroglitazar 4 mg/day/ Placebo/ 12 weeks for efficacy and 24 weeks for safety | **Efficacy**<br>Primary outcome:<br>Percentage reduction in TG<br>Secondary outcome:<br>Change in lipid profile: apolipoprotein A1, Apo B, HDL-C, non-HDL-C, LDL-C, total cholesterol, and very LDL-C<br>Exploratory efficacy outcome: FPG<br>**Safety**<br>Clinical severity, Haematological variation, LFT, RFT, Other routine test, Myopathy, inflammatory and cardiovascular markers markers | A significant reduction of mean plasma triglyceride levels with 2mg and 4mg of saroglitazar compared with placebo.<br>**Saroglitazar**<br>2 mg significantly decreased the levels of non-HDL-C, very LDL-C, total cholesterol, and fasting plasma glucose<br>Saroglitazar 4mg significantly reduced LDL-C and apolipoprotein B levels<br>**Safety**<br>No significant difference in interventional and control arm | This double blind RCT reported a beneficial effect of saroglitazar in improving the hypertriglyceridemia in patients with T2DM |

Apo B: Apolipoprotein B; dL: Decilitre; FPG: Fasting Plasma Glucose; HDL-C: High Density lipoprotein-Cholesterol; LDL-C: Low Density lipoprotein-Cholesterol; mg: Milligram; RCT: Randomized Controlled Trial; SR: Sustained Release; T2DM: Type-2-diabtes mellitus.

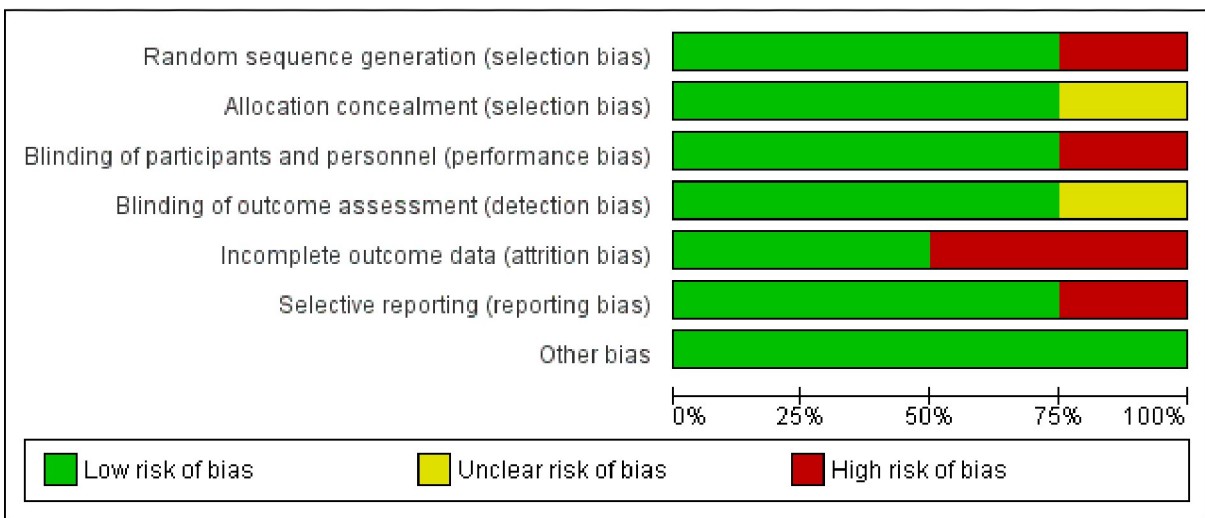

**Fig 2. Risk of bias graph.** Review author's judgements about each risk of bias assessment presented as percentages across all included studies.

control in patients with diabetes-related dyslipidemia (Table 3). No statistically significant heterogeneity was observed among included studies (P = 0.42; $I^2$ = 0%).

**LDL-cholesterol.** The pooled analysis of two studies[12,19] compared the saroglitazar 4 mg with saroglitazar 2mg, where 138 patients in each group resulted in an SMD of -0.24 **mg/ dL,** (95% CI -0.47 to 0.00), P = 0.05 (Fig 6). This demonstrates a significant decrease in the LDL cholesterol among the Saroglitazar 4 mg/day group compared to saroglitazar 2 mg/day group in patients with diabetes-related dyslipidemia (Table 3). There was no statistically significant heterogeneity among included studies (p = 0.44; $I^2$ = 0%). At the same time, three studies compared the saroglitazar 4 mg with control, where there were 148 patients in the saroglitazar 4 mg/day group and 144 patients in the control group[12,18,19]. The results showed an SMD of -0.36 **mg/dL,** (95% CI -0.59 to -0.12), p = 0.0026 (Fig 7). This demonstrates a significant decrease in the LDL cholesterol among the saroglitazar 4 mg/day group compared to the control in patients with diabetes-related dyslipidemia (Table 3). No statistically significant heterogeneity was detected among included studies (P<0.83; $I^2$ = 0%).

**Triglycerides.** The pooled analysis of two studies [12,19] compared the saroglitazar 4 mg with Saroglitazar 2mg, where there were 138 patients in each group, resulting in an SMD of -0.24 **mg/dL,** (95% CI -0.64 to 0.15), P = 0.22 (Table 3). There was no statistically significant heterogeneity among included studies (P = 0.13; $I^2$ = 55%). This demonstrates a non-significant decrease in the triglycerides among the saroglitazar 4 mg/day group compared to the saroglitazar 2 mg/day group in patients with diabetes-related dyslipidemia (Table 3). Besides, four studies compared the saroglitazar 4 mg with control, where there were 167 patients in the saroglitazar 4 mg/day group and 162 patients in the control group [12,18,19,23]. The results showed an SMD of -0.28 **mg/dL,** (95% CI -0.84 to 0.27), P = 0.31 (Table 3). This demonstrates a decrease in the triglycerides among the saroglitazar 4 mg/day group compared to the control group in patients with diabetes-related dyslipidemia (Table 3). No statistically significant heterogeneity was observed among included studies (P<0.01; $I^2$ = 78%).

**HDL-cholesterol.** The pooled analysis of two studies [12,19] compared the saroglitazar 4 mg with Saroglitazar 2mg, where there were 138 patients in each group. The results showed an SMD of -0.17 **mg/dL,** (95% CI -0.41 to 0.07), P = 0.15 (Table 3). This demonstrates a non-significant decrease in the HDL-cholesterol among the Saroglitazar 4 mg/day group compared to the saroglitazar 2 mg/day group in patients with diabetes-related dyslipidemia (Table 3). No

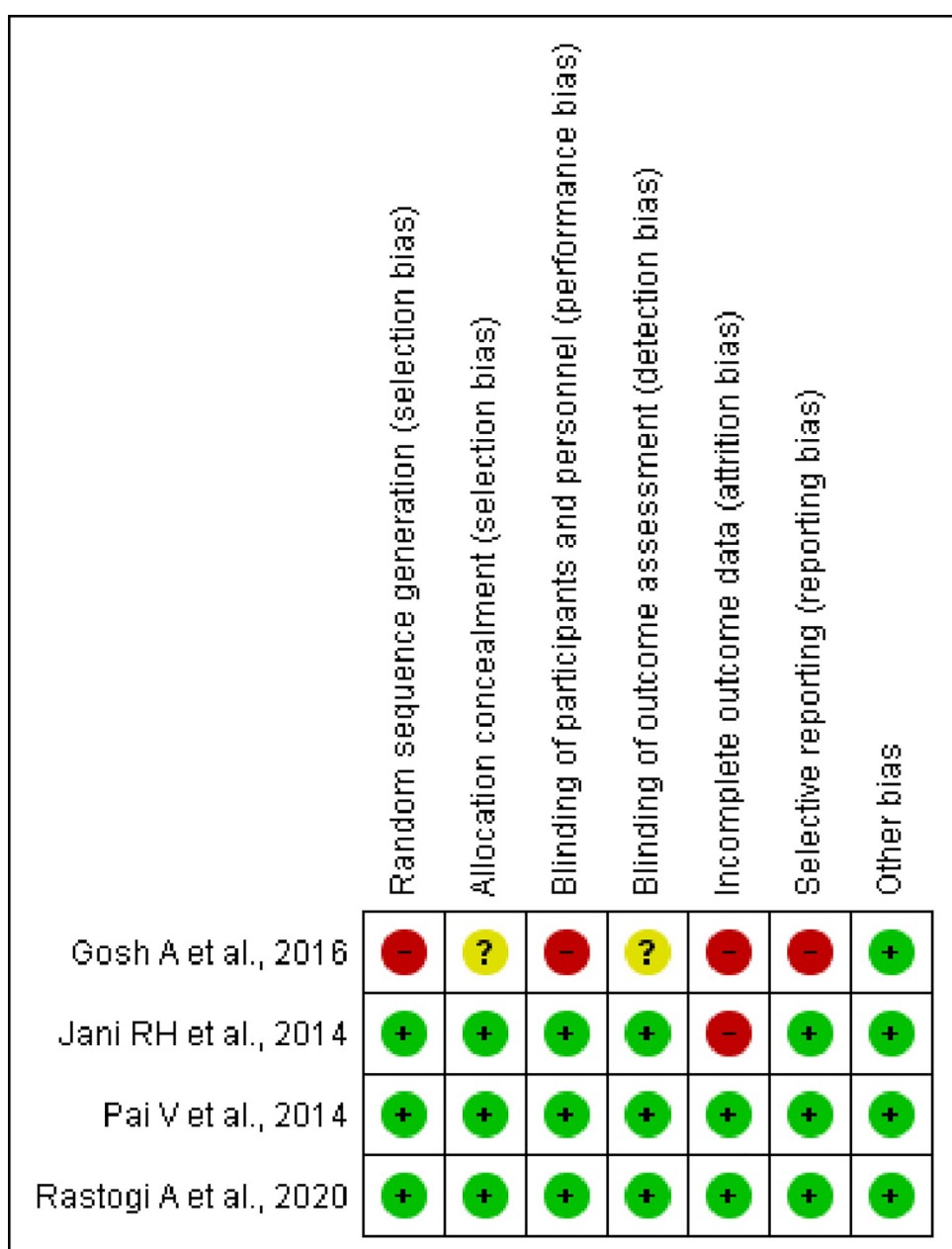

**Fig 3. Risk of bias summary.** Review author's judgements about each risk of bias item for each included study.

statistically significant heterogeneity was observed among included studies (P = 0.62; $I^2$ = 0%). Besides, three studies [12,18,19] compared the saroglitazar 4 mg/day with control, where there were 148 patients in saroglitazar 4 mg/day group and 144 patients in the control group. The results showed an SMD of 0.24 **mg/dL,** (95% CI -0.29 to 0.76), P = 0.37 (Table 3). This demonstrates a non-significant decrease in the HDL-cholesterol among the saroglitazar 4 mg/day group compared to the control in patients with diabetes-related dyslipidemia (Table 3). Statistically, significant heterogeneity was observed among included studies (P = 0.04; $I^2$ = 70%).

**Fasting plasma glucose.**   Two studies [12,19] compared the saroglitazar 4 mg with Saroglitazar 2mg, where there were 138 patients in each group. The results showed an SMD of -0.07

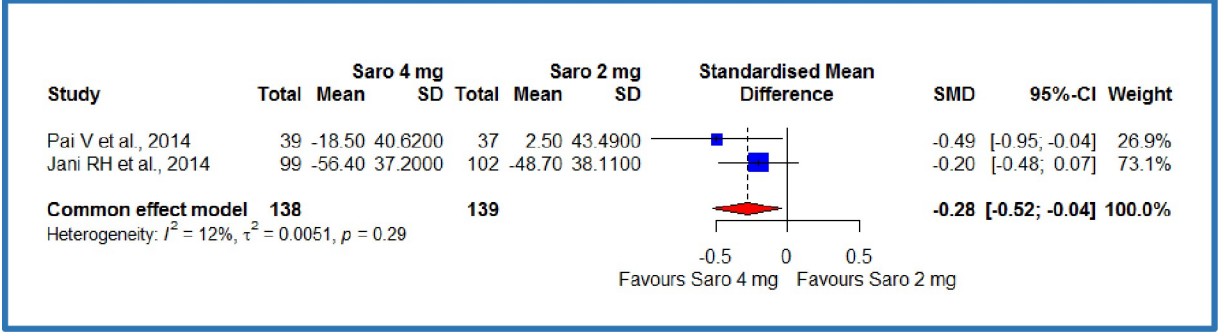

**Fig 4. Forest plot of comparison of saroglitazar 4mg versus saroglitazar 2mg on total cholesterol (mg/dL).**

**mg/dL,** (95% CI -0.30 to 0.17), P = 0.56 (Table 3). This demonstrates a non-significant decrease in the fasting plasma glucose among the saroglitazar 4 mg/day group compared to Saroglitazar 2 mg/day group in patients with diabetes-related dyslipidemia (Table 3). No

**Table 3. Mean absolute change results of all clinical outcomes measured in diabetes related dyslipidemia.**

| Clinical Outcomes | No of studies | Test for heterogeneity | | | Test of association | | | |
|---|---|---|---|---|---|---|---|---|
| | | Q | P | I² (%) | SMD | 95% CI | Z | P |
| **Triglycerides (mg/dL)** | | | | | | | | |
| Saroglitazar 4mg vs Control | 4 | 13.37 | <0.01 | 78 | -0.28 | -0.84; 0.27 | -1.00 | 0.31 |
| Saroglitazar 4mg vs 2mg | 2 | 2.24 | 0.13 | 55 | -0.24 | -0.64; -0.15 | -1.22 | 0.22 |
| **Total Cholesterol (mg/dL)** | | | | | | | | |
| Saroglitazar 4mg vs Control | 3 | 1.73 | 0.42 | 0 | -0.49 | -0.72; -0.26 | -4.12 | **<0.0001** |
| Saroglitazar 4mg vs 2mg | 2 | 1.14 | 0.29 | 12 | -0.28 | -0.52; -0.04 | -2.33 | **0.01** |
| **LDL-C (mg/dL)** | | | | | | | | |
| Saroglitazar 4mg vs Control | 3 | 0.38 | 0.83 | 0 | -0.36 | -0.59; -0.12 | -3.01 | **0.0026** |
| Saroglitazar 4mg vs 2mg | 2 | 0.59 | 0.44 | 0 | -0.23 | -0.47; -0.00 | -1.96 | **0.05** |
| **HDL-C (mg/dL)** | | | | | | | | |
| Saroglitazar 4mg vs Control | 3 | 6.58 | 0.04 | 70 | 0.24 | -0.29; 0.76 | 0.89 | 0.37 |
| Saroglitazar 4mg vs 2mg | 2 | 0.24 | 0.62 | 0 | -0.17 | -0.41; 0.07 | -1.41 | 0.15 |
| **Non-HDL-C (mg/dL)** | | | | | | | | |
| Saroglitazar 4mg vs Control | 3 | 15.19 | <0.01 | 87 | -0.66 | -1.77; 0.45 | -1.17 | 0.24 |
| **FPG (mg/dL)** | | | | | | | | |
| Saroglitazar 4mg vs Control | 4 | 48.03 | <0.01 | 94 | -0.22 | -1.38; 0.94 | -0.36 | 0.71 |
| Saroglitazar 4mg vs 2mg | 2 | 0.38 | 0.53 | 0 | -0.07 | -0.30; 0.17 | -0.57 | 0.56 |
| **Serum Creatinine (mg/dL)** | | | | | | | | |
| Saroglitazar 4mg vs Control | 2 | 12.29 | <0.01 | 92 | 0.58 | -0.43; 1.59 | 1.13 | 0.25 |
| Saroglitazar 4mg vs 2mg | 2 | 0.69 | 0.40 | 0 | 0.45 | 0.21; 0.69 | 3.64 | **0.0003** |
| **Alanine Amino transferase (U/L)** | | | | | | | | |
| Saroglitazar 4mg vs Control | 2 | 0.68 | 0.41 | 0 | -0.17 | -0.51; 0.05 | -1.40 | 0.16 |
| Saroglitazar 4mg vs 2mg | 2 | 1.18 | 0.27 | 15 | -0.07 | -0.31; 0.16 | -0.58 | 0.56 |
| **Aspartate Amino transferase (U/L)** | | | | | | | | |
| Saroglitazar 4mg vs Control | 2 | 0.78 | 0.38 | 0 | 0.05 | -0.18; 0.28 | 0.41 | 0.68 |
| Saroglitazar 4mg vs 2mg | 2 | 0 | 0.54 | 0 | -0.06 | -0.29; 0.18 | -0.47 | 0.64 |
| **Wieght Change (kg)** | | | | | | | | |
| Saroglitazar 4mg vs Control | 2 | 8.94 | <0.01 | 89 | -0.09 | -0.92; 0.73 | -0.22 | 0.82 |
| Saroglitazar 4mg vs 2mg | 2 | 0.26 | 0.61 | 0 | 0.27 | 0.03; 0.51 | 2.18 | **0.02** |

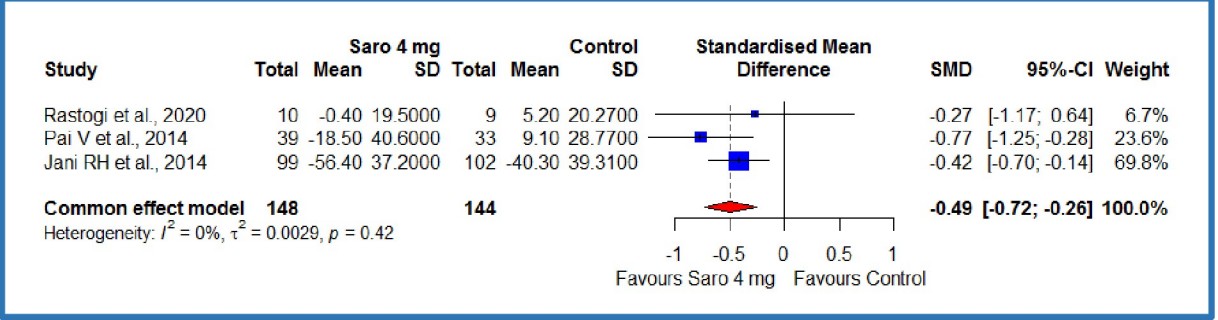

**Fig 5. Forest plot of comparison of saroglitazar 4mg versus control on total cholesterol (mg/dL).**

statistically significant heterogeneity was observed among included studies (P = 0.54; $I^2$ = 0%). Besides, four studies [12,18,19,22] compared the saroglitazar 4 mg with control, where there were 167 patients in saroglitazar 4 mg/day group and 162 patients in the control group. The results showed an SMD of -0.22 **mg/dL,** (95% CI -1.38 to 0.94), P = 0.71 (Table 3). This demonstrates a non-significant decrease in the fasting plasma glucose among the saroglitazar 4 mg/day group compared to the control in patients with diabetes-related dyslipidemia (Table 3). Statistically, significant heterogeneity was observed among included studies (P = 0.01; $I^2$ = 94%).

**Non-HDL cholesterol.** Three studies [18,19,22] compared the saroglitazar 4 mg/day with control, where there were 68 patients in the saroglitazar 4 mg/day group and 60 patients in the control group. The results showed a SMD of -0.66 **mg/dL,** (95% CI -1.77 to 0.45), P = 0.24 (Table 3). It demonstrates a non-significant decrease in the non-HDL cholesterol among the saroglitazar 4 mg/day group compared to the control in patients with diabetes-related dyslipidemia (Table 3). Statistically, significant heterogeneity was observed among included studies (P<0.01; $I^2$ = 87%).

## Safety of saroglitazar

**Serum creatinine levels.** Two studies [12,19] compared the saroglitazar 4 mg with the saroglitazar 2mg, where there were 132 patients in the saroglitazar 4 mg/day group, and 138 patients in the saroglitazar 2 mg/day group. The meta-analysis demonstrated a significant increase (SMD: 0.45 mg/dL; 95% CI 0.21 to 0.69; P = 0.0003) [Fig 8] in the serum creatinine levels among the saroglitazar 4 mg/day group compared to the saroglitazar 2 mg/day group in patients with diabetes-related dyslipidemia (Table 3). No statistically significant heterogeneity was observed among included studies (P = 0.41; $I^2$ = 0%). Besides, two studies [12,19] compared the saroglitazar 4 mg with control, where there were 132 patients in the saroglitazar 4

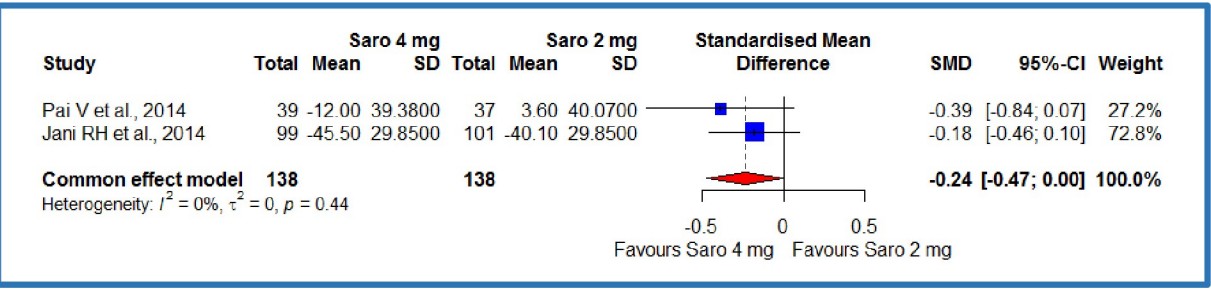

**Fig 6. Forest plot of comparison of saroglitazar 4mg versus saroglitazar 2mg on LDL-cholesterol (mg/dL).**

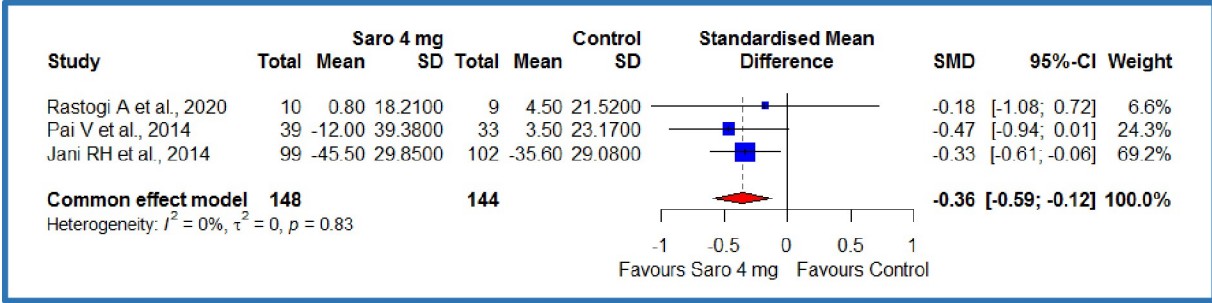

**Fig 7. Forest plot of comparison of saroglitazar 4mg versus control on LDL-cholesterol (mg/dL).**

mg/day group and 139 patients in the control group. The results showed a SMD of 0.58 **mg/dL,** (95% CI -0.43 to 1.59), P = 0.25 (Table 3). It demonstrates a non-significant increase in the serum creatinine levels among the saroglitazar 4 mg/day group compared to the control in patients with diabetes-related dyslipidemia (Table 3). Statistically significant heterogeneity was observed among included studies (P<0.01; $I^2$ = 92%).

## Bodyweight reduction

Two studies [12,19] compared the saroglitazar 4 mg with saroglitazar 2mg, where there were 132 patients in saroglitazar 4 mg/day group and 138 patients in Saroglitazar 2 mg/day group. The results showed an SMD of 0.27 **kg,** (95% CI 0.03 to 0.51), P = 0.02 (Table 3). It demonstrates a significant increase in the bodyweight reduction among the saroglitazar 4 mg/day group compared to the saroglitazar 2 mg/day group in patients with diabetes-related dyslipidemia (Table 3). No statistically significant heterogeneity was observed among included studies (P = 0.61; $I^2$ = 0%). Besides, two studies [12,19] compared the saroglitazar 4 mg with control, where there were 132 patients in the saroglitazar 4 mg/day group and 139 patients in the control group. The results showed an SMD of -0.09 **kg,** (95% CI -0.92 to 0.73), P = 0.82 (Table 3). It demonstrates a non-significant decrease in the bodyweight reduction among the saroglitazar 4 mg/day group compared to the control in patients with diabetes-related dyslipidemia (Table 3). Statistically, significant heterogeneity was observed among included studies (P<0.01; $I^2$ = 89%).

## Alanine aminotransferase

Two studies [12,19] compared the saroglitazar 4 mg with saroglitazar 2mg, where there were 132 patients in Saroglitazar 4 mg/day group and 138 patients in saroglitazar 2 mg/day group.

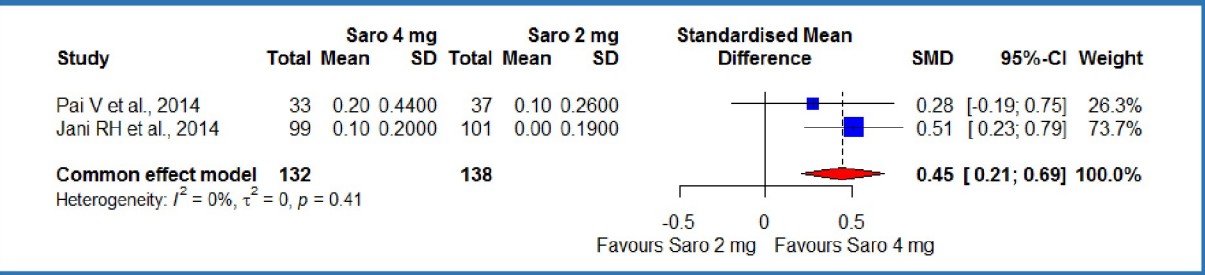

**Fig 8. Forest plot of comparison of saroglitazar 4mg versus saroglitazar 2 mg on serum creatinine (mg/dL).**

The results showed an SMD of -0.07 **U/L,** (95% CI -0.31 to 0.17), P = 0.56 (Table 3) which demonstrates a non-significant decrease in the alanine aminotransferase among the Saroglitazar 4 mg/day group compared to the saroglitazar 2 mg/day group in patients with diabetes-related dyslipidemia (Table 3). There was no statistically significant heterogeneity among included studies (P = 0.28; $I^2$ = 15%). Besides, two studies [12,19] compared the saroglitazar 4 mg with control, where there were 132 patients in the saroglitazar 4 mg/day group and 139 patients in the control group. The results showed an SMD of -0.17 **U/L,** (95% CI -0.41 to 0.07), P = 0.16 (Table 3). This demonstrates a non-significant decrease in the alanine aminotransferase among the saroglitazar 4 mg/day group compared to the control in patients with diabetes-related dyslipidemia (Table 3). No statistically significant heterogeneity was observed among included studies (P = 0.41; $I^2$ = 0%).

### Aspartate aminotransferase

Two studies [12,19] compared the saroglitazar 4 mg with saroglitazar 2mg, where there were 132 patients in the saroglitazar 4 mg/day group and 138 patients in Saroglitazar 2 mg/day group. The results showed an SMD of -0.06 **U/L,** (95% CI -0.30 to 0.18), P = 0.64 (Table 3). This demonstrates a non-significant decrease in the aspartate aminotransferase among the saroglitazar 4 mg/day group compared to saroglitazar 2 mg/day group in patients with diabetes-related dyslipidemia (Table 3). No statistically significant heterogeneity was observed among included studies (P = 0.55; $I^2$ = 0%). Besides, two studies [12,19] compared the Saroglitazar 4 mg with control, where there were 132 patients in Saroglitazar 4 mg/day group and 139 patients in the control group. The results showed an SMD of -0.05 **U/L,** (95% CI -0.19 to 0.29), P = 0.68 (Table 3). This demonstrates a non-significant decrease in the aspartate aminotransferase among the saroglitazar 4 mg/day group compared to the control in patients with diabetes-related dyslipidemia (Table 3). No statistically significant heterogeneity was observed among included studies (P = 0.318; $I^2$ = 0%). The summary of all clinical outcomes are presented in Table 3.

## Discussion

The current systematic review and meta-analysis looked at the efficacy and safety of Saroglitazar among patients with diabetic dyslipidemia with or without diabetes. This study is the first of its kind that attempted to summarise the efficacy and safety profiles of Saroglitazar; the overall results of this study are encouraging with a reduction in fasting plasma glucose, triglycerides, total cholesterol, LDL-cholesterol, serum creatinine, and body weight, along with improved safety outcomes including hepatotoxicity and renal toxicity.

Diabetic dyslipidemia is a condition that is characterized by low concentrations of HDL with elevated triglycerides, LDL, and postprandial lipemia [24]. At the same time, hypertriglyceridemia is also a notable risk factor and is present in almost 50% of the patients with type-2 diabetes, which is often unresponsive to statins [25]. Since higher levels of triglycerides could significantly increase the risk of cardiovascular diseases, triglycerides management has become the chief priority, especially in patients with diabetes [26]. saroglitazar is a dual peroxisome proliferator-activated receptor (PPAR) agonist with predominant PPAR-α and moderate -γ agonism that can optimise both lipid and glycaemic levels.

Among patients with diabetic dyslipidemia who were not responded to the atorvastatin therapy, a phase III study of saroglitazar has shown a significant improvement in triglyceride, LDL cholesterol, non-HDL-C VLDL, total cholesterol, and fasting plasma glucose [12]. The findings of another phase III clinical trial inferred that saroglitazar could be an alternative for patients those experience the common side effects of fibrates and pioglitazone. However, there

is a further need for studies of saroglitazar to well-establish its role in the management of diabetes with dyslipidemia [27]. Furthermore, alongside the favourable finding from the existing clinical trials [12,18–22], the results of a post-marketing study had demonstrated a significant reduction in the triglycerides (35.8%), non-HDL-C (23.4%), LDL-C (16.4%), and total cholesterol (19%) including HbA1c, FBG and PPBG levels with the saroglitazar 4 mg in patients with diabetic dyslipidemia [28].

Overall, when compared to control, once-daily dose of saroglitazar ranging from 2 to 4mg for a duration of 12 to 24 weeks has shown a standard mean difference of -0.28, -0.49, -0.36, 0.24, -0.66, 0.58, -0.09, -0.17, and 0.05, on the triglycerides, total cholesterol, LDL-cholesterol, HDL-cholesterol, non-HDL-cholesterol, serum creatinine, body weight, ALT, and AST, respectively. Furthermore, 4mg saroglitazar has predominantly shown better results when compared with the 2mg Saroglitazar, where a standard mean difference of -0.24, -0.28, -0.23, -0.17, 0.45, -0.07, -0.06, and 0.27, on the triglycerides, total cholesterol, LDL-cholesterol, HDL-cholesterol, serum creatinine, body weight, ALT, and AST, respectively.

In addition, 4 mg saroglitazar has shown the potentiality in decreasing glycaemic-related outcome measures such as FPG to a significant extent (SMD of -0.22 and -0.07) when compared to the control and 2 mg saroglitazar [19,21], which might be a result of its PPAR-c agonist activity. As a result, saroglitazar is emerging as a first-line choice for diabetic dyslipidemia with hypertriglyceridemia for its both lipid and glucose-lowering effects. Safety analysis of saroglitazar revealed no serious adverse events, including hepatotoxicity and renal toxicity in any of the clinical trials, which might be due to its non-renal route of excretion.[29] Instead, it showed a reduction in the ALT (absolute weighted mean difference of 13.6), AST (13.9) and serum creatinine levels for saroglitazar 2mg (0.51) [19,21]. All the reported adverse events like dyspepsia, gastritis, pyrexia were mild in severity and ranged from 10% to 17%, which seems to be better compared to the control groups (up to 20%) [19,20].

Besides exhibiting improved outcomes among patients with diabetes, saroglitazar also showed a significant reduction in serum triglyceride and VLDL levels and increased HDL levels in patients with HIV-associated lipodystrophy [20]. Among patients who are unresponsive or contra-indicated to fibrates and statin therapies, saroglitazar has the potential to undertake the cardiovascular risk associated with high triglyceridemia [19]. Also, since saroglitazar has both antidyslipidemic and anti glycemicglycaemic actions, it would have the ability to address the uncertainties in the reduction of macrovascular and microvascular complications. However, as most of these studies are clinical trials with shorter durations, findings on more extensive observational studies using real-world evidence are warranted.

## Limitations

There are certain limitations associated with this study that must be considered while interpreting the findings. Firstly, all the studies included in this systematic review and meta-analysis were clinical trials, which usually account for a limited sample size with a minimum duration of the follow-up period, which makes it difficult to assess the long-term safety and efficacy of saroglitazar. Secondly, some of these studies were single-center and single-arm trials without a comparator group. Thirdly, these findings could not be generalizable to all the patients with dyslipidemia, as all the included studies were solely conducted in India. In addition to this, all the included trials had shorter follow-up duration with surrogate outcomes.

## Conclusions

The current evidences, suggests saroglitazar to be effective in reducing LDL cholesterol, non-HDL-C, triglycerides, total cholesterol, and fasting plasma glucose among the dyslipidemic

patients. Furthermore, it is appeared to be safe in terms of liver enzymes abnormality and body weight. However, studies are needed to compare the effect of saroglitazar with other anti-hyperlipidemic medications.

## Supporting information

**S1 Checklist.**
(DOC)

**S1 Table. Risk of bias analysis of single arm studies.**
(DOCX)

**S1 File. Search strategy for each database.**
(DOCX)

**S2 File. Risk of bias analysis of single arm studies.**
(DOCX)

## Author Contributions

**Conceptualization:** Manik Chhabra, Kota Vidyasagar, Sai Krishna Gudi, Jatin Sharma, Rishabh Sharma, Muhammed Rashid.

**Data curation:** Manik Chhabra, Kota Vidyasagar, Sai Krishna Gudi, Jatin Sharma, Rishabh Sharma, Muhammed Rashid.

**Formal analysis:** Manik Chhabra, Kota Vidyasagar, Sai Krishna Gudi, Jatin Sharma, Rishabh Sharma, Muhammed Rashid.

**Investigation:** Manik Chhabra, Sai Krishna Gudi, Jatin Sharma, Muhammed Rashid.

**Methodology:** Manik Chhabra, Sai Krishna Gudi, Jatin Sharma, Muhammed Rashid.

**Project administration:** Manik Chhabra, Sai Krishna Gudi, Muhammed Rashid.

**Software:** Sai Krishna Gudi, Muhammed Rashid.

**Supervision:** Manik Chhabra, Sai Krishna Gudi, Muhammed Rashid.

**Validation:** Manik Chhabra, Kota Vidyasagar, Rishabh Sharma, Muhammed Rashid.

**Visualization:** Manik Chhabra, Kota Vidyasagar, Jatin Sharma, Rishabh Sharma.

**Writing – original draft:** Manik Chhabra, Kota Vidyasagar, Jatin Sharma, Rishabh Sharma, Muhammed Rashid.

**Writing – review & editing:** Manik Chhabra, Muhammed Rashid.

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
