## [Decision Letter · Decision Letter 0]

26 Oct 2021

PONE-D-21-19447Efficacy and safety of Saroglitazar for the management of Dyslipidaemia: A systematic review and meta-analysis of interventional studiesPLOS ONE

Dear Dr. Rashid,

Thank you for submitting your manuscript to PLOS ONE. After careful consideration, we feel that it has merit but does not fully meet PLOS ONE’s publication criteria as it currently stands. Therefore, we invite you to submit a revised version of the manuscript that addresses the points raised during the review process. The reviewers' comments and suggestions have been attached below. Please carefully address these comments and suggestions and revise accordingly.

We look forward to receiving your revised manuscript.

Kind regards,

Jie V Zhao

Academic Editor

PLOS ONE

3. Please update search and analysis to include relevant studies published since May 2020.

Reviewers' comments:

Reviewer's Responses to Questions

**Comments to the Author**

1. Is the manuscript technically sound, and do the data support the conclusions?

Reviewer #1: Partly

Reviewer #2: Yes

2. Has the statistical analysis been performed appropriately and rigorously? 

Reviewer #1: I Don't Know

Reviewer #2: Yes

3. Have the authors made all data underlying the findings in their manuscript fully available?

Reviewer #1: Yes

Reviewer #2: Yes

4. Is the manuscript presented in an intelligible fashion and written in standard English?

Reviewer #1: Yes

Reviewer #2: Yes

5. Review Comments to the Author

Reviewer #1: In this systematic review with meta-analysis, the authors synthesize the evidence for saroglitazar on surrogate lipid outcomes (triglycerides, total cholesterol, LDL cholesterol, HDL cholesterol, non-HDL cholesterol) and safety outcomes (glucose, ALT, AST, and weight change). The author state that they use the PICOS framework to answer questions regarding efficacy and safety of saroglitazar, but it is unclear to me if the PICOS questions (and study protocol) were pre-specified prior to conducting the review.

Major comments

1. The study selection criteria could be more clearly explained. It isn’t clear to me if the patients in the eligible trials had to have diabetes or pre-diabetes or if they just needed to be adults > 18 years of age. Please clarify and add the % of patients with diabetes at baseline to your study summary Table 1.

2. A major problem is in the available underlying studies and the comparisons that were tested. Are the authors most interested in the comparative efficacy and safety of saroglitazar to placebo, or direct head-to-head comparisons to other PPAR agonists (e.g. fenofibrate, pioglitazone) or both? To best answer this question I suggest the author’s only include the highest quality evidence for assessing efficacy which would be a randomized clinical trial. Two studies (Bhosle 2018, Deshpande, and Pay 2014 ) should be excluded as they have no comparator (only treat with saroglitazar) so no conclusions can be drawn.

3. Outcomes: Please list your efficacy and safety outcomes in the methods. The most important outcomes (perhaps 3-5) should be the focus of the review. Your most important safety outcomes should also be included in the abstract.

4. Statistical methods: The authors need to specify the actual analytic package used in R software (R studio is the software ‘container’ used to run R and its packages). The authors need to provide more details about the type of meta-analysis method used.

5. Efficacy and effectiveness have different definitions. I believe the authors are assessing the efficacy of saroglitazar in the context of well-controlled trials.

6. I suggest avoiding using the terms statistically significant or non-significant decrease (page 10, line 218 for example) since it does adequately describe the strength and magnitude of the evidence. Some have suggested wording such as “there no evidence of a difference” or “there was evidence of a difference”. Stating there was a non-significant decrease is probably not appropriate since we do not truly know if there was a decrease or if the change is simply due to biological or analytic variability in the test.

7. Discussion: Please add a citation for the statement regarding first-line choice for dyslipidemia (p 17, line 320). The study didn’t assess glycosylated haemoglobin but it is listed as an outcome on line 322. Also the authors state that serum creatine was decreased (line 361) but in fact in was increase in the meta-analysis.

8. Figure 1 PRISMA: The number included in the quantitative analysis is incorrect. This should be the number of studies actually meta-analyzed. Please state the exact number included in the meta-analysis in the figure and the results section.

9. Conclusions: I don’t think the size and the quality of the included studies permits such strong conclusions in the discussion and abstract. The authors should avoid over interpreting their findings and should discuss the many limitations of the available evidence (e.g., small sample sizes, surrogate outcomes, short duration of follow-up etc.).

Minor comments

1. Introduction and abstract: It appears that saroglitazar is only approved for use in India. It would be helpful to put its approval and contemporary use in context. Are there any data on uptake or how it is used when compared with other PPAR agents?

2. Suggest further language editing. Please ensure consistent capitalization with journal guidelines for article title and text (e.g., saroglitazar, dyslipidemia).

3. Table 1, last row, column “Study design” should likely state for safety (for efficacy is repeated). I also suggest that there are different columns for the intervention and the comparator, rather than having this information in the “number of participants” column.

4. Table 2: There are symbols without a footnote (e.g., ‡). This table should be condensed and similar terminology for each study should be used (it appears that the text might be verbatim from the study publication). The authors should include the results (numeric) for the primary study efficacy and safety outcomes and not interpret the findings in the key conclusion column.

5. Figures: Suggest having forest plot labels “Favours saroglitazar” and “Favours placebo” or the other comparator.

Reviewer #2: In this meta-analysis, the authors investigate the efficacy and safety of Saroglitazar in patients with dyslipidaemia. The results indicate that Saroglitazar has beneficial effects on improving dyslipidaemia in patients with type-2 diabetes. This study has practical and clinical value. Overall, the methods seem sound and the conclusions make a significant contribution to the field. However, in my opinion, this paper also has some important issues that need to be addressed by the authors.

First, a more precise description of the methods should be provided (potentially as supplementary material). For instance:

* Can the authors provide more information on the inclusion and exclusion criteria? “The highly irrelevant studies only were excluded during the title/abstract screening.” What are the reasons and rationale for the irrelevance? And more information can be added to the PRISMA flow diagram, i.e., the number of excluded reviews, observational studies, animal studies, etc. should be illustrated separately.

* It is said that “we conducted the systematic review and meta-analysis aimed to examine the effectiveness and safety of Saroglitazar for the management of dyslipidaemia.” How was this effectiveness and safety defined and what was the measurements of them? There should be a description of these aspects in the method section.

* Description of statistical analysis is missing.

As for the result section, the description of each part of the results should be more varied, otherwise it will make the paper less readable. There are also some very obvious errors in the reference citations and tables/figures. For instance:

* “Two studies [20, 21] compared the Saroglitazar 4 mg with Saroglitazar 2mg”, these two studies should be by Pai V et al., 2014 and Jani RH et al., 2014, not by Deshpande A et al., 2016 and Bhosle D et al., 2018. It is important to double check all the reference citations and not to make such mistakes.

* In the table 1, the number of participants in Pai V et al., 2014 should be “Pioglitazon (45mg): 40”, not “Saroglitazor (45mg): 40”. Such mistakes should be avoided.

And the results of the two single-arm studies are not described.

More cautious formulations could be used in several cases. For instance:

* In the discussion (and the conclusion), it is claimed that “The overall results of this study are encouraging with a significant reduction in glycosylated haemoglobin, fasting plasma glucose, triglycerides, total cholesterol, LDL-cholesterol, serum creatinine ". This conclusion seems misleading given the non-significance of the result (and the existence of high heterogeneity; as acknowledged by the authors in the result session).

* It should be clear from the result of the abstract that this study has examined the safety of Saroglitazar.

* Reference citations should be added in some place, e.g., “which might be due to its non-renal route of excretion”.

Finally, it is true that because of the limited number of studies and sample size, this study has significant limitations and the robustness of the conclusions is not sufficient. I suggest to include a paper published recently by Krishnappa, M et al. which included 1155 patients examining the effectiveness of saroglitazar 2mg and 4mg on glycemic control, lipid profile in patients with type 2 diabetes. This study may strengthen the reliability of meta-analysis.

Ref: Krishnappa M, Patil K, Parmar K, Trivedi P, Mody N, Shah C, Faldu K, Maroo S; PRESS XII study group, Parmar D. Effect of saroglitazar 2 mg and 4 mg on glycemic control, lipid profile and cardiovascular disease risk in patients with type 2 diabetes mellitus: a 56-week, randomized, double blind, phase 3 study (PRESS XII study). Cardiovasc Diabetol. 2020 Jun 19;19(1):93. doi: 10.1186/s12933-020-01073-w. PMID: 32560724; PMCID: PMC7305598.

6. PLOS authors have the option to publish the peer review history of their article (what does this mean?). If published, this will include your full peer review and any attached files.

Reviewer #1: **Yes: **Joseph E Blais

Reviewer #2: **Yes: **HUANG Xin

---

## [Author Response · Author response to Decision Letter 0]

22 Feb 2022

Respected Editor-In-Chief

Thank you for your timely review of our manuscript entitled “Efficacy and safety of Saroglitazar for the management of Dyslipidemia: A systematic review and meta-analysis of interventional studies” [PONE-D-21-19447]. We have gone through all of your valuable comments. Please find the revisions and justifications made based on comment to comment.

Reviewers' comments:

Reviewer #1

In this systematic review with meta-analysis, the authors synthesize the evidence for saroglitazar on surrogate lipid outcomes (triglycerides, total cholesterol, LDL cholesterol, HDL cholesterol, non-HDL cholesterol) and safety outcomes (glucose, ALT, AST, and weight change). The author state that they use the PICOS framework to answer questions regarding efficacy and safety of saroglitazar, but it is unclear to me if the PICOS questions (and study protocol) were pre-specified prior to conducting the review.

Response#

We have used PICOS framework and it was pre-specified prior to the conduct of the search for the electronic databases. It is well explained in the methods context of the manuscript. Unfortunately, we did not register our protocol with PROSPERO, because of COVID-19, it was a delay. 

Major comments

Comment 1#

The study selection criteria could be more clearly explained. It isn’t clear to me if the patients in the eligible trials had to have diabetes or pre-diabetes or if they just needed to be adults > 18 years of age. Please clarify and add the % of patients with diabetes at baseline to your study summary Table 1.

Response#

For the clarity of respected reviewer, we have undertaken a systematic literature review to study the effect of Saroglitazar on the lipid profile irrespective of diabetes. The concern of the reviewer is clarified in the context of the methods section and it is re-written as described below for a better clarity.

“The research question for our systematic review is "What is the safety and efficacy of Saroglitazar for the management of hypercholesterolemia?" The research question was broken down into PICOS (population, intervention, comparison, outcome, and study design) format. Population (P) comprises of adult patients with hypercholesterolemia with or without diabetes. Intervention (I) consists of Saroglitazar in any of its effective dose. Comparator (C) were any comparator such as different doses, placebo or other drug as per author’s discretion. Outcomes (O) considered were efficacy and safety of Saroglitazar. The effect of Saroglitazar on total cholesterol, LDL-cholesterol, triglycerides, HDL-cholesterol, non-HDL cholesterol, and fasting plasma glucose considered to be efficacy outcomes. While reduction in Serum Creatinine, Alanine Transaminases, Aspartate Transaminases, and body weight considered to be safety outcomes. In study design (S), we included all type of interventional studies; randomised, and non-randomised trials. Any studies satisfying the above specified PICOS were included in the systematic review. Reviews, observational and descriptive studies, editorials, commentaries and conference proceedings were excluded.”

Comment 2#

A major problem is in the available underlying studies and the comparisons that were tested. Are the authors most interested in the comparative efficacy and safety of saroglitazar to placebo, or direct head-to-head comparisons to other PPAR agonists (e.g. fenofibrate, pioglitazone) or both? To best answer this question I suggest the author’s only include the highest quality evidence for assessing efficacy which would be a randomized clinical trial. Two studies (Bhosle 2018, Deshpande, and Pay 2014) should be excluded as they have no comparator (only treat with saroglitazar) so no conclusions can be drawn.

Response#

Firstly, if we are excluding the single arm studies, we will be biased on evidence and our main objective is to provide very first evidence on efficacy and safety of Saroglitazar. Our aim is not to do any comparative efficacy, instead to give an overall picture on efficacy of Saroglitazar.

Secondly, if we consider your request and exclude two studies we may lose bulk in evidence. Moreover, the evidences derived from these studies are useful to compare the efficacy and safety in different disease of the drug. Furthermore, our safety evidence can be used as future research direction to conduct studies on real-world evidence of Saroglitazar.

Comment 3#

Outcomes: Please list your efficacy and safety outcomes in the methods. The most important outcomes (perhaps 3-5) should be the focus of the review. Your most important safety outcomes should also be included in the abstract.

Response

Thanks for highlighting the flaw, we have modified it accordingly.

“Outcomes (O) considered were efficacy and safety of Saroglitazar. The effect of Saroglitazar on total cholesterol, LDL-cholesterol, triglycerides, HDL-cholesterol, non-HDL cholesterol, and fasting plasma glucose considered to be efficacy outcomes. While reduction in Serum Creatinine, Alanine Transaminases, Aspartate Transaminases, and body weight considered to be safety outcomes.”

We have added following sentence to abstract

“Saroglitazar showed a reduction in liver transaminases and serum creatinine.”

4. Statistical methods: The authors need to specify the actual analytic package used in R software (R studio is the software ‘container’ used to run R and its packages). The authors need to provide more details about the type of meta-analysis method used.

Response#

Please find the addition with respect to the above comment

“All the analyses were performed with R software (R version 4.1.2) using meta-package. Changes in continuous outcomes were calculated for every included study arm by subtracting the value at baseline from the value after intervention. All the efficacy estimates were expressed as standardised mean difference (SMD) or absolute weighted mean change (MRAW) and 95% confidence interval (CI) from baseline. Standard deviations (SD) were calculated from the standard error or 95% CI, according to the Cochrane handbook for systematic review of interventions. The Higgins I2 statistics and Cochran’s Q test were used to assess the potential statistical heterogeneity among trials. The meta-analysis was conducted using a fixed-effect model (using inverse-variance) or a random-effect model (DerSimonian–Laird method) based on low heterogeneity (<50%) or high heterogeneity (>50%). Due to less number of included studies (<10), it was not feasible to examine the publication bias through funnel plot by Egger's and Begg’s test.” 

Comment 5#

 Efficacy and effectiveness have different definitions. I believe the authors are assessing the efficacy of saroglitazar in the context of well-controlled trials.

Response

Thank for the valuable suggestion, we have replaced word “effectiveness” with “efficacy”

Comment 6#

I suggest avoiding using the terms statistically significant or non-significant decrease (page 10, line 218 for example) since it does adequately describe the strength and magnitude of the evidence. Some have suggested wording such as “there no evidence of a difference” or “there was evidence of a difference”. Stating there was a non-significant decrease is probably not appropriate since we do not truly know if there was a decrease or if the change is simply due to biological or analytic variability in the test.

Response

Thank you for suggestion and we have considered your request and change made in the context is as follow 

“This demonstrates decrease in the triglycerides among the Saroglitazar 4 mg/day group compared to the control group in patients with”

Comment 7# 

Discussion: Please add a citation for the statement regarding first-line choice for dyslipidemia (p 17, line 320). The study didn’t assess glycosylated haemoglobin but it is listed as an outcome on line 322. Also the authors state that serum creatine was decreased (line 361) but in fact in was increase in the meta-analysis.

Response#

We have modified that sentence as it was bit contradictory, it is not first line of therapy. There is paucity of the literature stating it is first-line of therapy. Thanks for the pick point regarding the glycosylated hemoglobin, we have rectified our sentence.

Serum creatinine levels for Saroglitazar 2mg were decreased. We have elaborated in the results section and have modified the sentence picking up your concern.

In figure 8 we have compared Saroglitazar 4mg vs 2mg, S creatinine was decreased with 2 mg as compared to 4 mg.

Comment 8#

 Figure 1 PRISMA: The number included in the quantitative analysis is incorrect. This should be the number of studies actually meta-analyzed. Please state the exact number included in the meta-analysis in the figure and the results section.

Response#

Thank you for your valuable suggestion, we replaced 6 studied with 4 studies in quantitative section of the (updated) PRISMA.

Comment 9#

Conclusions: I don’t think the size and the quality of the included studies permits such strong conclusions in the discussion and abstract. The authors should avoid over interpreting their findings and should discuss the many limitations of the available evidence (e.g., small sample sizes, surrogate outcomes, short duration of follow-up etc.).

Response#

We have made changes as per your suggestion, the limitation section has been updated and changes made as follows

“There are certain limitations associated with this study that must be considered while interpreting the findings. Firstly, all the studies that are included in this systematic review & meta-analysis were clinical trials, which usually account for a limited sample size with a minimum duration of the follow-up period, which makes it difficult to assess the long-term safety and efficacy of Saroglitazar. Secondly, some of these studies were the single centre and single-arm trails without a comparator group. Thirdly, these findings could not be generalizable to all the patients with dyslipidemia, as all the included studies were solely conducted in India. In addition to this all the included trials were having shorter follow up duration with surrogate outcomes.”

Minor comments

Comment 1# 

Introduction and abstract: It appears that Saroglitazar is only approved for use in India. It would be helpful to put its approval and contemporary use in context. Are there any data on uptake or how it is used when compared with other PPAR agents?

Response#

We have added information on its approval and contemporary use in India as

“the treatment of diabetic dyslipidemia and hypertriglyceridemia” however, we didn’t get any data on its uptake and its use as compared to other PPAR analogues.

Comment2#

Suggest further language editing. Please ensure consistent capitalization with journal guidelines for article title and text (e.g., saroglitazar, dyslipidemia).

Response#

Our manuscript have undergone for English language proof reading by a native English speaker to avoid any flaws in English. We have modified concerns raised by the author

Comment 3#

Table 1, last row, column “Study design” should likely state for safety (for efficacy is repeated). I also suggest that there are different columns for the intervention and the comparator, rather than having this information in the “number of participants” column.

Response#

Thanks for your keen observation, we have made changes and it is now reflected in the last row of the column, it was typo error. Deleting arms from number of participants column will not be beneficial, as in current representation it gives clear picture of the number of participants over eyes.

Comment 4#

Table 2: There are symbols without a footnote (e.g., ‡). This table should be condensed and similar terminology for each study should be used (it appears that the text might be verbatim from the study publication). The authors should include the results (numeric) for the primary study efficacy and safety outcomes and not interpret the findings in the key conclusion column.

Response#

Table 2 is completely revised as per the reviewers comment

Comment 5#

Figures: Suggest having forest plot labels “Favours saroglitazar” and “Favours placebo” or the other comparator.

Response#

We have modified as suggested by the reviewer 

Reviewer #2: 

In this meta-analysis, the authors investigate the efficacy and safety of Saroglitazar in patients with dyslipidemia. The results indicate that Saroglitazar has beneficial effects on improving dyslipidemia in patients with type-2 diabetes. This study has practical and clinical value. Overall, the methods seem sound and the conclusions make a significant contribution to the field. However, in my opinion, this paper also has some important issues that need to be addressed by the authors.

First, a more precise description of the methods should be provided (potentially as supplementary material). For instance:

* Can the authors provide more information on the inclusion and exclusion criteria? “The highly irrelevant studies only were excluded during the title/abstract screening.” What are the reasons and rationale for the irrelevance? And more information can be added to the PRISMA flow diagram, i.e., the number of excluded reviews, observational studies, animal studies, etc. should be illustrated separately.

Response: Thank you for the valuable suggestion, we have updated the PRISMA flow diagram as suggested and we have added the information.

Comment# 

* It is said that “we conducted the systematic review and meta-analysis aimed to examine the effectiveness and safety of Saroglitazar for the management of dyslipidaemia.” How was this effectiveness and safety defined and what was the measurements of them? There should be a description of these aspects in the method section.

Response#

We wrongly used the term effectiveness considering the comments of reviewer one we have replaced with safety and efficacy. This is further addressed in the PICOS

Comment#

* Description of statistical analysis is missing.

Response#

All the analyses were performed with R software (R version 4.1.2) using meta-package. Changes in continuous outcomes were calculated for every included study arm by subtracting the value at baseline from the value after intervention. All the efficacy estimates were expressed as standardised mean difference (SMD) or absolute weighted mean change (MRAW) and 95% confidence interval (CI) from baseline. Standard deviations (SD) were calculated from the standard error or 95% CI, according to the Cochrane handbook for systematic review of interventions. The Higgins I2 statistics and Cochran’s Q test were used to assess the potential statistical heterogeneity among trials. The meta-analysis was conducted using a fixed-effect model (using inverse-variance) or a random-effect model (DerSimonian–Laird method) based on low heterogeneity (<50%) or high heterogeneity (>50%). Due to less number of included studies (<10), it is not feasible to examine the publication bias through funnel plot by Egger's and Begg’s test.” 

Comment#

As for the result section, the description of each part of the results should be more varied, otherwise it will make the paper less readable. There are also some very obvious errors in the reference citations and tables/figures. For instance:

* “Two studies [20, 21] compared the Saroglitazar 4 mg with Saroglitazar 2mg”, these two studies should be by Pai V et al., 2014 and Jani RH et al., 2014, not by Deshpande A et al., 2016 and Bhosle D et al., 2018. It is important to double check all the reference citations and not to make such mistakes.

Response#

We are really very sorry for such a critical error. It has been modified and taken care in the modified manuscript.

Comment#

* In the table 1, the number of participants in Pai V et al., 2014 should be “Pioglitazon (45mg): 40”, not “Saroglitazor (45mg): 40”. Such mistakes should be avoided.

And the results of the two single-arm studies are not described.

Response#

We have made changes and addition as per reviewer comment and it can be seen in the context

Comment#

More cautious formulations could be used in several cases. For instance:

* In the discussion (and the conclusion), it is claimed that “The overall results of this study are encouraging with a significant reduction in fasting plasma glucose, triglycerides, total cholesterol, LDL-cholesterol, serum creatinine ". This conclusion seems misleading given the non-significance of the result (and the existence of high heterogeneity; as acknowledged by the authors in the result session).

Response#

We have modified the whole manuscript as per the actual findings by avoiding these types of claims 

Comment# 

* It should be clear from the result of the abstract that this study has examined the safety of Saroglitazar.

Response#

Our objective is clearly states that, this study aimed “To evaluate the efficacy and safety profiles of Saroglitazar in patients with dyslipidemia”. However, we couldn’t specify all PICOS in detail in the abstract due to word limits. 

Further, we have added the statement with respect to the safety outcome in the abstract as follows:

“Saroglitazar showed a reduction in liver transaminases and serum creatinine” which considered to be safety outcomes in our study.

Comment#

* Reference citations should be added in some place, e.g., “which might be due to its non-renal route of excretion”.

Response#

Thanks for picky point, we have added reference

Comment#

Finally, it is true that because of the limited number of studies and sample size, this study has significant limitations and the robustness of the conclusions is not sufficient. I suggest to include a paper published recently by Krishnappa, M et al. which included 1155 patients examining the effectiveness of saroglitazar 2mg and 4mg on glycemic control, lipid profile in patients with type 2 diabetes. This study may strengthen the reliability of meta-analysis.

Ref: Krishnappa M, Patil K, Parmar K, Trivedi P, Mody N, Shah C, Faldu K, Maroo S; PRESS XII study group, Parmar D. Effect of saroglitazar 2 mg and 4 mg on glycemic control, lipid profile and cardiovascular disease risk in patients with type 2 diabetes mellitus: a 56-week, randomized, double blind, phase 3 study (PRESS XII study). Cardiovasc Diabetol. 2020 Jun 19;19(1):93. doi: 10.1186/s12933-020-01073-w. PMID: 32560724; PMCID: PMC7305598.

#Response

Thank you for suggesting the reference. However, we couldn’t include this study as per our review criteria. Our inclusion criteria clearly indicates the inclusion of “participants who diagnosed with hypercholesterolemia with or without diabetes”. Unfortunately this study included “population with diabetes with or without hypercholesterolemia”, which is out of our focus.

---

## [Decision Letter · Decision Letter 1]

15 Mar 2022

PONE-D-21-19447R1Efficacy and safety of Saroglitazar for the management of Dyslipidaemia: A systematic review and meta-analysis of interventional studiesPLOS ONE

Dear Dr. Muhammed,

Thank you for submitting your manuscript to PLOS ONE. After careful consideration, we feel that it has merit but does not fully meet PLOS ONE’s publication criteria as it currently stands. Therefore, we invite you to submit a revised version of the manuscript that addresses the points raised during the review process. Please refer to the comments raised by the reviewers and revise accordingly.

We look forward to receiving your revised manuscript.

Kind regards,

Jie V Zhao

Academic Editor

PLOS ONE

Reviewers' comments:

Reviewer's Responses to Questions

**Comments to the Author**

1. If the authors have adequately addressed your comments raised in a previous round of review and you feel that this manuscript is now acceptable for publication, you may indicate that here to bypass the “Comments to the Author” section, enter your conflict of interest statement in the “Confidential to Editor” section, and submit your "Accept" recommendation.

Reviewer #1: (No Response)

Reviewer #2: (No Response)

2. Is the manuscript technically sound, and do the data support the conclusions?

Reviewer #1: Partly

Reviewer #2: Yes

3. Has the statistical analysis been performed appropriately and rigorously? 

Reviewer #1: Yes

Reviewer #2: Yes

4. Have the authors made all data underlying the findings in their manuscript fully available?

Reviewer #1: Yes

Reviewer #2: Yes

5. Is the manuscript presented in an intelligible fashion and written in standard English?

Reviewer #1: Yes

Reviewer #2: Yes

6. Review Comments to the Author

Reviewer #1: Dr Chhabra and colleagues have conducted a systematic review of interventional studies for saroglitazar and included 4 studies in their meta-analysis. I appreciate the authors efforts to respond to the first round of comments and I can see they have made significant revisions to strengthen their manuscript. However, I still have several concerns regarding the methods and presentation of the results. My major concern is about the latest date of the literature search which was performed 22 months ago (May 2020).

1. Have the authors updated their search of the literature? The date of the last search was May 2020 which is not very recent. I will like to see an updated literature search in case recently published studies have been missed.

2. Abstract, results: Total cholesterol and LDL cholesterol, saraglitazar 4 mg vs control does not match the number of studies included in the results in the text and (Figures 5 and 7 only have 3 studies). The sentence (line 67) to the abstract “Saroglitazar showed a reduction in liver transaminases and serum creatinine” is too vague. I suggest you add the same details as the earlier sentences (intervention, comparator, estimate, p value # of studies).

3. P values for estimates should not be dichotomized at 0.05 level. Please include the actual p value for the pooled estimates in the abstract and results.

4. Please add the unit of measurement for lipids (mmol/L) be added to the forest plot figure titles and Table 3.

5. There is a discrepancy between figure 1 tiff file and figure 1 docx file in the number of studies included in meta-analysis.

Language issues:

Abbreviations: Sometimes LDL cholesterol and LDL-cholesterol (recommend not using the dash as this seems to be the norm for most publications).

Line 2, line 51 and others: No need to specifically capitalize saroglitazar and dyslipidemia

Line 431: use and not &

Line 435: misspelled trials as trails.

Throughout the paper the authors should consistently use the term dyslipidemia which, to my understanding, encompasses any problematic concentration of blood lipid measures such as hyperlipidemia, hypertriglyceridemia or low HDL cholesterol.

Reviewer #2: Thank you for the revision. The majority of concerns have been addressed. Before the manuscript is acceptable for publication, I think a few issues need to be revised again.

First, some more useful information should be added to the abstract section. For the method, it is not necessary to state “Two individual reviewers performed an independent review for study selection, data extraction and quality assessment.”. A summary description of the selection of outcomes or the method of data synthesis can be provided. And for the results part, it is not comprehensive to describe only the significant results. All study results should be summarized because you have discussed them in the discussion section.

Secondly, I agree with reviewer 1 that avoiding using the terms statistically significant or non-significant decrease (not only for TG). Except for TC and LDL-C, the P-values for other results are greater than 0.05, we do not know if there was a truly decrease. And in the discussion and conclusion part, “the overall results of this study are encouraging with a reduction in fasting plasma glucose, triglycerides, total cholesterol, LDL-cholesterol,” such concluding statement exaggerates your results, the significance and strength of the evidence do not support such a conclusion. Perhaps you can say that such results (TG, FPG) could be considered suggestive.

Thirdly, I don’t agree the statement that Saroglitazar showed a reduction in liver transaminases, serum creatinine and body weight (abstract line 67 and discussion line 379), as the results for Saroglitazar 4mg vs control were not significant and the results for 4mg vs control and 4mg vs 2mg were reversed for body weight and Aspartate Amino transferase. It can only be said that there is no evidence that Saroglitazar has effects on these safety outcomes. And the SMD for serum creatinine were positive, which is inconsistent with what is stated in the abstract and discussion.

7. PLOS authors have the option to publish the peer review history of their article (what does this mean?). If published, this will include your full peer review and any attached files.

Reviewer #1: **Yes: **Joseph Blais

Reviewer #2: **Yes: **Xin Huang

---

## [Author Response · Author response to Decision Letter 1]

4 Apr 2022

Respected Editor-In-Chief

Thank you for your timely review of our manuscript entitled “Efficacy and safety of Saroglitazar for the management of Dyslipidemia: A systematic review and meta-analysis of interventional studies” [PONE-D-21-19447]. We have gone through all of your valuable comments. Please find the revisions and justifications made based on comment to comment in this rebuttal letter and the same has been updated in the revised manuscript (R2) with track changes.

Reviewer #1: Dr Chhabra and colleagues have conducted a systematic review of interventional studies for saroglitazar and included 4 studies in their meta-analysis. I appreciate the authors efforts to respond to the first round of comments and I can see they have made significant revisions to strengthen their manuscript. However, I still have several concerns regarding the methods and presentation of the results. My major concern is about the latest date of the literature search which was performed 22 months ago (May 2020).

Comment 1#

Have the authors updated their search of the literature? The date of the last search was May 2020 which is not very recent. I will like to see an updated literature search in case recently published studies have been missed.

Response#

Yes, we have updated the literature search in January 2022 and we could identify a single new study related to Saroglitazar, unfortunately, that study didn’t meet our inclusion criteria.

We have updated the same in our methodology part which is as follow 

“The initial search was performed in April 2019 and was updated in January 2022.”

Comment 2#

Abstract, results: Total cholesterol and LDL cholesterol, saraglitazar 4 mg vs control does not match the number of studies included in the results in the text and (Figures 5 and 7 only have 3 studies). 

The sentence (line 67) to the abstract “Saroglitazar showed a reduction in liver transaminases and serum creatinine” is too vague. I suggest you add the same details as the earlier sentences (intervention, comparator, estimate, p value # of studies).

Response#

Thanks for the picky point we have rectified number of studies from 4 to 3

“and control [SMD: −0.36 mmol/L, 95% CI −0.59 to -0.12; p < 0.01; 3 studies].”

“[SMD − 0.49 mmol/L, 95% CI: − 0.72 to -0.26; p < 0.0001; 3 studies].”

Considering you suggestion about the vague sentence we have removed it from the abstract and instead of the there was requirement of the sentence, we kept new sentence as “Saroglitazar was not associated with any serious adverse events.”

Comment 3. P values for estimates should not be dichotomized at 0.05 level. Please include the actual p value for the pooled estimates in the abstract and results.

Response#

Thanks for your suggestion. As you suggested, we have replaced dichotomized P-values to exact P-Values in both abstract and the results section.

Comment 4. Please add the unit of measurement for lipids (mmol/L) be added to the forest plot figure titles and Table 3.

Response#

Thanks for your suggestion. As you suggested, we have added the unit of measurement in the forest plot figure titles, Table 3, abstract, and the text in the results section for each and every included outcome 

Comment 5. There is a discrepancy between figure 1 tiff file and figure 1 docx file in the number of studies included in meta-analysis.

Response#

This was happened as we updated Figure 1 during revision and we accidently forgot remove the tiff file submitted during the initial submission. Now we have attached the final figure 1 (R2) and removed the old file from the submission

Comment 6

Language issues:

Abbreviations: Sometimes LDL cholesterol and LDL-cholesterol (recommend not using the dash as this seems to be the norm for most publications).

Response#

Rectified as per your suggestion, Thanks

Comment 7

Line 2, line 51 and others: No need to specifically capitalize saroglitazar and dyslipidemia

Response#

We have considered your suggestion and changes made are in whole context of the manuscript can be seen as track on mode

Comment 8

Line 431: use and not &

Response#

Rectified, thanks for your suggestion

Comment 9

Line 435: misspelled trials as trails.

Response#

Rectified, thanks for your suggestion

Comment 10

Throughout the paper the authors should consistently use the term dyslipidemia which, to my understanding, encompasses any problematic concentration of blood lipid measures such as hyperlipidemia, hypertriglyceridemia or low HDL cholesterol.

Response#

Thanks for the suggestion, we have rectified it

Reviewer #2:

Thank you for the revision. The majority of concerns have been addressed. Before the manuscript is acceptable for publication, I think a few issues need to be revised again.

Comment 1#

First, some more useful information should be added to the abstract section. For the method, it is not necessary to state “Two individual reviewers performed an independent review for study selection, data extraction and quality assessment.”. A summary description of the selection of outcomes or the method of data synthesis can be provided. And for the results part, it is not comprehensive to describe only the significant results. All study results should be summarized because you have discussed them in the discussion section.

Response#

We have considered your suggestion and we have added the outcomes considered and details on quality assessment in the methodology part of the abstract which is as follows:

“The efficacy of saroglitazar was assessed with respect to its effect in total cholesterol, LDL and HDL-cholesterol, triglycerides, fasting plasma glucose, and non-HDL cholesterol. The effects on serum creatinine levels, bodyweight reduction, alanine aminotransferase and aspartate aminotransferase were considered to be safety endpoint. The Cochrane risk of bias assessment tool was used to assess the methodological quality of the included studies”

Comment 2#

Secondly, I agree with reviewer 1 that avoiding using the terms statistically significant or non-significant decrease (not only for TG). Except for TC and LDL-C, the P-values for other results are greater than 0.05, we do not know if there was a truly decrease. And in the discussion and conclusion part, “the overall results of this study are encouraging with a reduction in fasting plasma glucose, triglycerides, total cholesterol, LDL-cholesterol,” such concluding statement exaggerates your results, the significance and strength of the evidence do not support such a conclusion. Perhaps you can say that such results (TG, FPG) could be considered suggestive.

Response#

We have modified such exaggerating results as per the reviewer’s comments

Comment 3#

Thirdly, I don’t agree the statement that Saroglitazar showed a reduction in liver transaminases, serum creatinine and body weight (abstract line 67 and discussion line 379), as the results for Saroglitazar 4mg vs control were not significant and the results for 4mg vs control and 4mg vs 2mg were reversed for body weight and Aspartate Amino transferase. It can only be said that there is no evidence that Saroglitazar has effects on these safety outcomes. And the SMD for serum creatinine were positive, which is inconsistent with what is stated in the abstract and discussion.

Response#

We have carefully gone through the statement and have removed the sentence from the abstract, Thanks for your kind suggestion.

7. PLOS authors have the option to publish the peer review history of their article (what does this mean?). If published, this will include your full peer review and any attached files.

Do you want your identity to be public for this peer review? For information about this choice, including consent withdrawal, please see our Privacy Policy.

Reviewer #1: Yes: Joseph Blais

Reviewer #2: Yes: Xin Huang

---

## [Decision Letter · Decision Letter 2]

25 Apr 2022

PONE-D-21-19447R2Efficacy and safety of saroglitazar for the management of dyslipidaemia: A systematic review and meta-analysis of interventional studiesPLOS ONE

Dear Dr. Muhammed,

Thank you for submitting your manuscript to PLOS ONE. After careful consideration, we feel that it has merit but does not fully meet PLOS ONE’s publication criteria as it currently stands. Therefore, we invite you to submit a revised version of the manuscript that addresses the points raised during the review process.

We look forward to receiving your revised manuscript.

Kind regards,

Jie V Zhao

Section Editor

PLOS ONE

Journal Requirements:

Reviewers' comments:

Reviewer's Responses to Questions

**Comments to the Author**

1. If the authors have adequately addressed your comments raised in a previous round of review and you feel that this manuscript is now acceptable for publication, you may indicate that here to bypass the “Comments to the Author” section, enter your conflict of interest statement in the “Confidential to Editor” section, and submit your "Accept" recommendation.

Reviewer #1: (No Response)

2. Is the manuscript technically sound, and do the data support the conclusions?

Reviewer #1: Yes

3. Has the statistical analysis been performed appropriately and rigorously? 

Reviewer #1: Yes

4. Have the authors made all data underlying the findings in their manuscript fully available?

Reviewer #1: Yes

5. Is the manuscript presented in an intelligible fashion and written in standard English?

Reviewer #1: No

6. Review Comments to the Author

Reviewer #1: Thank you for responding to the previous comments and updating the literature search. I still have some minor comments that are directed at strengthening the manuscript for publication.

Abstract

Objective, second sentence (line 51): I suggest you revise the second sentence to "This systematic review aimed to evaluate...".

You should be consistent with abbreviations, such as LDL cholesterol. First use of the term should be spelled out, then subsequently abbreviated.

Results, last sentence (line 69): Serious adverse events were not an outcome of this study. Please revise to state safety endpoints or list the specific outcomes were no association was observed.

Main paper: Conclusion (line 441)

VLDL particles were not assessed as an outcome in this review. Please change to triglycerides as this is the surrogate measure for VLDL that you assessed in the study.

Presentation

Since PLOS ONE does not copyedit papers, I strongly suggest that you have someone external to the authors proofread the paper to make sure the next version has a consistent format and has no spelling or grammatical errors.

Two issues I noticed:

- There should be a space between dose and unit for saroglitazar (e.g, 2 mg not 2mg).

- Spelling of weight in Table 3 is wrong and the clean (no tracked changes) version seems to be missing the units for each outcome.

I don't believe the "Review criteria" and "Message for the clinic" sections are required for PLOS ONE. Please check the submission guidelines.

7. PLOS authors have the option to publish the peer review history of their article (what does this mean?). If published, this will include your full peer review and any attached files.

Reviewer #1: **Yes: **Joseph Blais

---

## [Author Response · Author response to Decision Letter 2]

11 May 2022

REBUTTAL LETTER-R4

Respected Editor-In-Chief

Thank you for your timely review of our manuscript entitled “Efficacy and safety of Saroglitazar for the management of Dyslipidemia: A systematic review and meta-analysis of interventional studies” [PONE-D-21-19447]. We have gone through all of your valuable comments. Please find the revisions and justifications made based on comment to comment in this rebuttal letter and the same has been updated in the revised manuscript (R3) with track changes.

Comment 1

Please ensure that you refer to Table 1 in your text as, if accepted, production will need this reference to link the reader to the Table.

Response#

Based on the reviewer comment, we referred Table 1 in the text before the appearance of table.

Comment 2

Please include your tables as part of your main manuscript and remove the individual files.

Response#

Based on the reviewer comment, we included tables as part of the manuscript and removed individual files.

Thanks for reviewing our paper and providing us with the comments to improve further

---

## [Editor Report · Decision Letter 3]

24 May 2022

Efficacy and safety of saroglitazar for the management of dyslipidaemia: A systematic review and meta-analysis of interventional studies

PONE-D-21-19447R3

Dear Dr. Muhammed,

We’re pleased to inform you that your manuscript has been judged scientifically suitable for publication and will be formally accepted for publication once it meets all outstanding technical requirements.

Kind regards,

Jie V Zhao

Section Editor

PLOS ONE
---

## [Editor Report · Acceptance letter]

9 Jun 2022

PONE-D-21-19447R3 

Efficacy and safety of saroglitazar for the management of dyslipidemia: A systematic review and meta-analysis of interventional studies 

Dear Dr. Rashid:

I'm pleased to inform you that your manuscript has been deemed suitable for publication in PLOS ONE. Congratulations! Your manuscript is now with our production department. 

Kind regards, 

on behalf of

Dr. Jie V Zhao 

Section Editor

PLOS ONE